# Transient destabilization of whole brain dynamics induced by N,N-Dimethyltryptamine (DMT)
Juan Ignacio Piccinini [1] ✉, Yonatan Sanz Perl [1,2], Carla Pallavicini[1,3], Gustavo Deco [2,4], Morten Kringelbach [5,6,7], David Nutt [8], Robin Carhart-Harris[8,9], Christopher Timmermann[8] & Enzo Tagliazucchi [1,10] ✉

The transition towards the brain state induced by psychedelic drugs is frequently neglected in favor of a static description of their acute effects. We use a time-dependent whole-brain model to reproduce large-scale brain dynamics measured with fMRI from 15 volunteers under 20 mg intravenous N,N-Dimethyltryptamine (DMT), a short-acting psychedelic. To capture its transient effects, we parametrize the proximity to a global bifurcation using a pharmacokinetic equation. Simulated perturbations reveal a transient of heightened reactivity concentrated in fronto-parietal regions and visual cortices, correlated with serotonin 5HT2a receptor density, the primary target of psychedelics. These advances suggest a mechanism to explain key features of the psychedelic state and also predicts that the temporal evolution of these features aligns with pharmacokinetics. Our results contribute to understanding how psychedelics introduce a transient where minimal perturbations can achieve a maximal effect, shedding light on how short psychedelic episodes may extend an overarching influence over time.

Psychedelic drugs offer an opportunity to investigate how changes in the brain across multiple spatiotemporal scales interact with human consciousness and cognition[1]. At the molecular level, psychedelics bind to the serotonin 5HT2a receptor[2], recruiting specific intracellular signaling pathways that are different from those implicated in the action of non-psychedelic 5HT2A agonists[3,4]. The subjective effects of psychedelics may also depend on other pharmacological and non-pharmacological factors[5], the latter including the context of drug intake and the mindset of the user[6]. At the systems level, psychedelics increase global network integration measured with functional magnetic resonance imaging (fMRI)[7–12], and evidence from multiple modalities links their effects to increased entropy and complexity of spontaneous brain activity fluctuations[13–19]. An integrative understanding of psychedelic action would require the identification of causal mechanisms behind these empirical observations, from molecules to subjective experience[1].

In recent years, generative whole-brain activity models have been increasingly adopted to test potential mechanisms underlying neuroimaging data, including successful applications to the specific case of psychedelic compounds psilocybin and lysergic acid diethylamide (LSD)[20]. In these two case studies, biophysical models consisting of local excitatory and inhibitory populations with excitatory long-range connections were used to provide evidence supporting psychedelic-induced modulation of 5HT2a synaptic scaling[21–23]. Herzog and colleagues implemented a similar model to show that 5HT2a receptor stimulation is consistent with increased brain-wide entropy[13], in agreement with the theoretical model of psychedelic action proposed by Carhart-Harris[24]. A complementary approach is to consider phenomenological models to investigate changes in global brain dynamics in line with insights and metrics from complexity science[25]. This approach was followed to demonstrate that LSD increases the complexity of spontaneous brain activity as assessed via fMRI[26], an observation consistent

[1]Universidad de Buenos Aires, Facultad de Ciencias Exactas y Naturales, Departamento de Física, and CONICET - Universidad de Buenos Aires, Instituto de Física Aplicada e Interdisciplinaria (INFINA), Buenos Aires, Argentina. [2]Center for Brain and Cognition, Computational Neuroscience Group, Department of Information and Communication Technologies, Universitat Pompeu Fabra, Barcelona, Spain. [3]Integrative Neuroscience and Cognition Center, CNRS, Université Paris Cité, Paris, France. [4]Institució Catalana de la Recerca i Estudis Avançats (ICREA), Barcelona, Spain. [5]Centre for Eudaimonia and Human Flourishing, University of Oxford, Oxford, UK. [6]Department of Psychiatry, University of Oxford, Oxford, UK. [7]Center for Music in the Brain, Department of Clinical Medicine, Aarhus University, Aarhus, Denmark. [8]Centre for Psychedelic Research, Department of Brain Sciences, Faculty of Medicine, Imperial College London, London, UK. [9]Psychedelics Division, Neuroscape, Department of Neurology, University of California San Francisco, San Francisco, CA, USA. [10]Latin American Brain Health Institute (BrainLat), Universidad Adolfo Ibañez, Santiago, Chile. ✉e-mail: piccijuan@gmail.com; tagliazucchi.enzo@googlemail.com

with previous studies showing an expanded repertoire of brain states with facilitated transitions between them[19,27], constituting a potential mechanism for the psychedelic-induced enhancement of neural flexibility[28]. Moreover, the dynamical characterization of psychedelic action reveals a scenario opposite to that of unconsciousness, where a reduction in the repertoire of brain states and an increase in their stability against perturbations is observed[25,29,30] as hypothesized in previous theoretical work[24].

To date, whole-brain models have been used to investigate the mechanisms underlying the steady-state effects of LSD and psilocybin, two classic psychedelic drugs. However, it is currently unknown whether the putative mechanisms of psychedelic action identified by these models also extend to the transitions from pre-dose baseline and the acute effects. Time-dependent models capable of fitting dynamics across transitions are necessary to distinguish between the neurochemical modulation of brain activity, aligned with the drug pharmacokinetics, and the indirect effects of factors linked to the interplay between expectations, subjective effects, and the short-term impact of the experience on the emotional state of the subjects[1]. Moreover, developing computational models sensitive to slow temporal fluctuations in brain activity may contribute to determine whether the pharmacology of certain psychedelics is multi-phasic, as has been proposed for the case of LSD[31]. The short-lasting psychedelic effects of intravenous DMT are ideally suited for this purpose, as they are highly dynamic and give way to the recovery of the baseline state after around 30 minutes[32]. Making progress in this direction, recent studies have found that specific EEG features of the DMT state are correlated with the serum concentration of the drug[17,33]. However, the dynamical principles behind the transition to altered consciousness remain to be addressed by modeling efforts.

In this work, we adopted a model previously used to investigate the effects of psychedelic drugs[26], introducing time dependency in the parameter governing the proximity to dynamical criticality, i.e. a phase transition between qualitatively different dynamics (chaos/noise vs. statistical regularities/oscillations[34]). To model the relatively short-lasting effects induced by DMT, the temporal evolution of the model bifurcation parameter was constrained to represent a simplified version of DMT pharmacokinetics[35]. The optimization of the free parameters underlying this temporal evolution allowed us to reproduce the empirical functional connectivity dynamics (FCD)[36], and subsequently the optimal peak intensity and latency for DMT vs. a placebo control condition. Finally, we investigated the stability of the simulated dynamics against external perturbations, thus assessing potential correlations between the local regional reactivity and the density of 5HT2a receptors, the main pharmacological target of DMT[2].

## Results

### Methodological overview

Figure 1 provides an outline of the methods and the overall procedure. Following previous work, the local dynamics in the whole-brain model were given by Stuart-Landau nonlinear oscillators, with a bifurcation at $a = 0$[36]. For $a > 0$ the model presents stable oscillations, while $a < 0$ results in stable spirals, which extinguish the oscillation amplitude until the dynamics are dominated by the additive noise term, $\eta(t)$. Close to the bifurcation (i.e. dynamic criticality), the noise introduces spontaneous switches between both regimes, resulting in oscillations with complex amplitude fluctuations. To model the time-dependent changes introduced by DMT, we put forward an assumed equation for the bifurcation parameter $a(t)$ given by the gamma function shown in Fig. 1. Parameters $\lambda$ and $\beta$ determine the peak amplitude and its latency, respectively. For an adequate choice of parameters, this function rises rapidly and then presents a slow decay, constituting an approximate description of the pharmacokinetics of bolus intravenous DMT administration[35]. The local Stuart-Landau nonlinear oscillators were coupled using an averaged structural connectivity matrix inferred using diffusion tensor imaging (DTI) from healthy individuals and scaled by the global coupling parameter $G$. Since we aimed to capture the temporal evolution introduced by the short-acting effects of DMT, we optimized the gamma function parameters $\lambda$ and $\beta$ to fit the reproduction of the mean FCD computed from $n = 15$ participants, which contains in its $i, j$ entry the

similarity between functional connectivity (FC) matrices computed over short windows starting at time points $i$ and $j$ (see Methods). Note that in contrast to previous applications of this model to resting state data[21,34,36–38], we did not optimize the statistical similarity between the empirical and simulated FCD matrices; instead, we employed the Euclidean distance -also known as Frobenius distance- for their comparison, as we were interested in capturing the temporal evolution introduced by DMT.

### Model optimization

To determine the optimal gamma function parameters to reproduce the whole-brain dynamics represented by the FCD, we performed an exhaustive exploration of all pairwise combinations of $\lambda$ and $\beta$ within a range of values compatible with the expected DMT pharmacokinetics (see Methods). The results of this exploration are shown in Fig. 2A for the placebo (left) and DMT conditions (right), with lower values indicating a better model fit (average of $n = 50$ independent simulations). For the placebo condition, the optimal values were located within a large region of comparatively low amplitude and variable latency, evidencing a weak temporal dependency without a clearly defined peak of maximal intensity. Conversely, for DMT this region was displaced and reduced to encompass shorter and less variable latencies, together with larger amplitude values. Figure 2B represents the empirical (left column) and optimal simulated (right column) FCD matrices for the placebo (first row) and DMT conditions (second row). Note that the two diagonal blocks of the FCD matrices separate the baseline (see Fig. 1, "Functional connectivity dynamics") and post-administration periods. The model is capable of approximating this temporal structure, as well as the overall intensity of the matrix element values.

After determining the optimal $\lambda$ and $\beta$ for each independent run of the simulation, we explored the corresponding $a(t)$ plots displayed in Fig. 3A for placebo and DMT, with thicker lines indicating the mean across all $n = 50$ simulations. This figure shows that dynamics start from a baseline of sustained oscillations at $a = 0.07$. After DMT infusion, we observed a sharp and rapid decrease of $a(t)$ which displaced whole-brain dynamics towards the bifurcation at $a = 0$. Reaching its peak after $\approx$ 5 minutes, the parameter $a(t)$ gradually recovered towards baseline values at the end of the scanning session. In contrast, results for the placebo condition showed a lower peak amplitude combined with longer latencies, to the point where the peaks were not reached during the scanning session. As a consequence, $a(t)$ for the placebo condition approximated the constant baseline value with comparatively smaller temporal variation. Figure 3B summarizes the differences between the dynamics of $a(t)$ found for placebo and DMT in the two-dimensional space spanned by $\lambda$ and $\beta$, where each point represents optimal values for an independent run of the simulation, and the larger full circles indicate the mean across all simulations. For the DMT condition, we could observe that $\lambda$ and $\beta$ were clustered in the upper left corner, indicating comparatively low latencies and high amplitude peaks ($\lambda = 159.3 \pm 7$, $\beta = 284 \pm 37$, 95% confidence interval [CI]). Conversely, the placebo condition resulted in low amplitude values and more variable latencies, skewed towards larger values relative to DMT ($\lambda = 65.6 \pm 9$, $\beta = 588 \pm 69$, 95% CI). Both clusters of values could be clearly separated, supporting the finding of qualitatively different $a(t)$ dynamics between conditions ($t$-test $p < 0.0001$ for both parameters).

### Oscillatory perturbations produce transients of heightened reactivity in DMT

After determining the optimal parameters to fit the FCD, we used the resulting models to investigate the time-dependent effects of delivering external perturbations. This analysis was motivated in previous reports indicating that psychedelics increase the reactivity to external stimuli and facilitate transitions between brain metastable states[26,27]. Based on these results, we hypothesized that DMT would result in a transient episode of more sensitive brain dynamics, arising due to a destabilization of whole-brain dynamics, i.e. due to the increased proximity to dynamic criticality.

To test our hypothesis, we introduced a periodic external perturbation at the frequency of the endogenous oscillations, which was previously shown

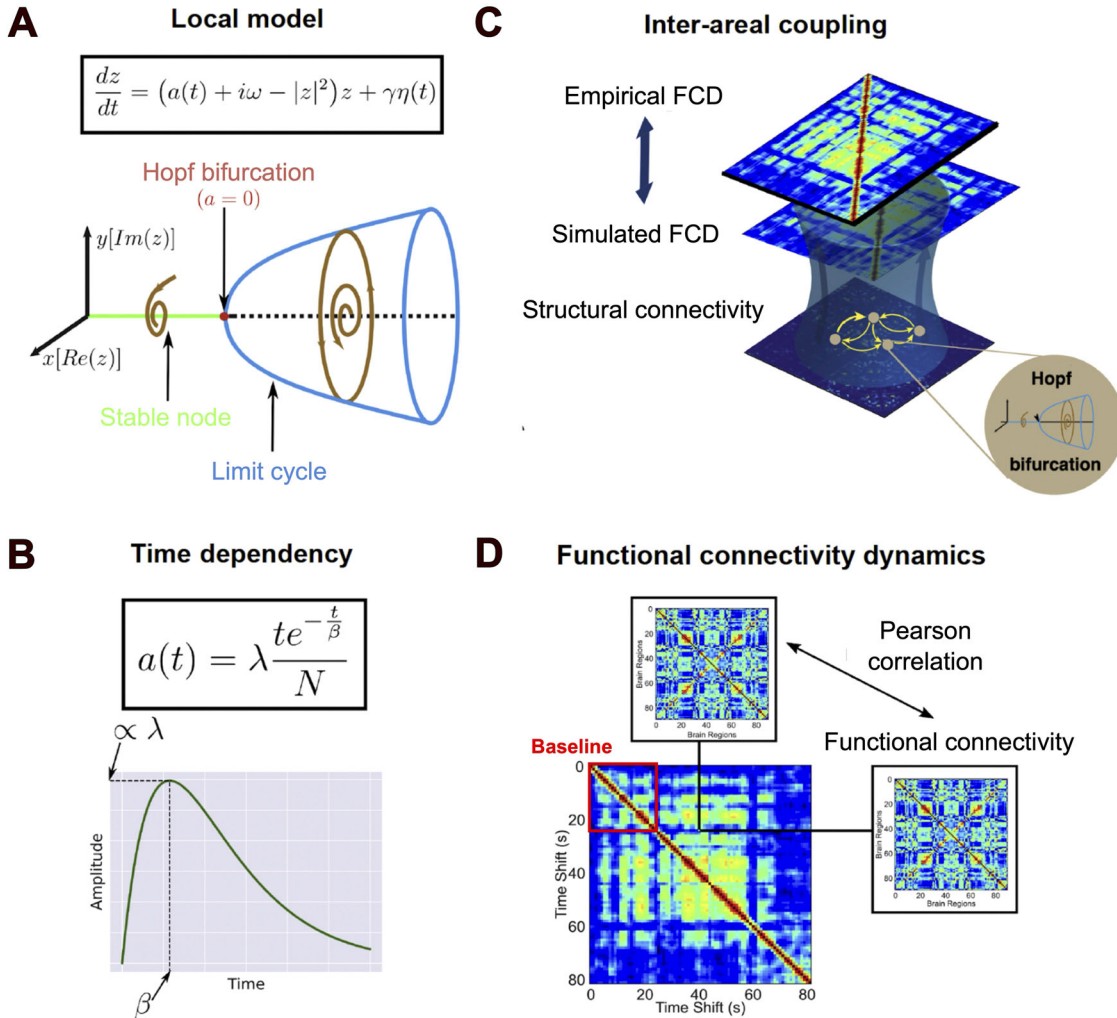

**Fig. 1 | Overview of the whole-brain model and the choice of optimization target.**
**A** Dynamics of a single node ("Local Model"), consisting of a Stuart-Landau non-linear oscillator with bifurcation parameter $a$, and the imaginary and real part of complex variable $z$ vs. $a$, showing a transition between stable spirals and limit cycles at $a = 0$. **B** Temporal parametrization of the bifurcation parameter $a(t)$, given by a gamma function with parameters $\lambda$ (peak amplitude) and $\beta$ (peak latency).
**C** Illustration of how the nodes were coupled ("Inter-areal coupling") following the structural connectivity given by DTI scaled by $G$ to reproduce the empirical FCD. **D** Representation of the computation of the FCD matrix, which contains in its $i, j$ entry the similarity between FC matrices computed over short windows starting at time points $i$ and $j$. The diagonal block encased in red indicates the baseline period before the administration of DMT (FCD: functional connectivity dynamics; DMT: dimethyltryptamine; DTI: diffusion tensor imaging).

to maximize the effect on the ongoing brain activity[39,40]. We determined the reactivity or sensitivity to the perturbation at each time point, noted here by $\chi(t)$, computing the derivative of the induced FCD changes with a numerical differentiation algorithm (see methods for more details, and see Supplementary Fig. 1 for an example of how FCD changes at two different stimulation intensities). To facilitate the interpretation of the results, we introduced this perturbation at nodes located within six resting state networks (RSN)[41] known to encompass different functional brain systems: primary visual (Vis), extrastriate (ES), auditory (Aud), sensorimotor (SM), default mode (DM) and executive control (EC) cortices. Figure 4A shows $\chi(t)$ for each RSN, both for the placebo (top) and DMT (bottom) conditions (see Supplementary Fig. 2 for an analysis of $\chi(t)$ computed using inter-network FCD only). For the latter, it is clear that $\chi(t)$ is aligned with the expected evolution of drug effect intensity, with the largest reactivity obtained at the extrastriate visual cortex. In contrast, the time dependency of $\chi(t)$ was considerably less marked for the placebo group. Figure 4B displays the peak of $\chi(t)$, i.e. $\chi_{max}$, for each RSN and for three different external perturbation intensities ($F_{ext}$), comparing the DMT and placebo conditions.

Consistent with the results shown in Fig. 4A, the peak values were systematically lower in placebo vs. DMT for all intensities, indicating higher reactivity to external perturbations during the acute effects of the drug.

## Correlation between peak differential reactivity and local 5HT2a receptor density

Finally, we investigated the correlation between the 5HT2a receptor density (see Methods) and the peak reactivity ($\chi_{max}$) across RSN. Based on the known pharmacological action of DMT[2], we expected a positive correlation between these two variables, i.e. that RSN with higher 5HT2a receptor density would show higher $\chi_{max}$ values, and vice-versa. Figure 5A illustrates the spatial configuration of the RSN, while the mean 5HT2a receptor density per RSN is shown in Fig. 5B. To assess if the difference in $\chi_{max}$ between conditions can be attributed to the local density of serotonin receptors, we first computed the difference between the reactivity curves of DMT and placebo, resulting in $\Delta\chi(t)$. Figure 5C shows the peak of $\Delta\chi(t)$, $\Delta\chi_{max}$, vs. the 5HT2a receptor density of each RSN, together with the best least-squares linear fit. To estimate the significance of this correlation, we conducted a

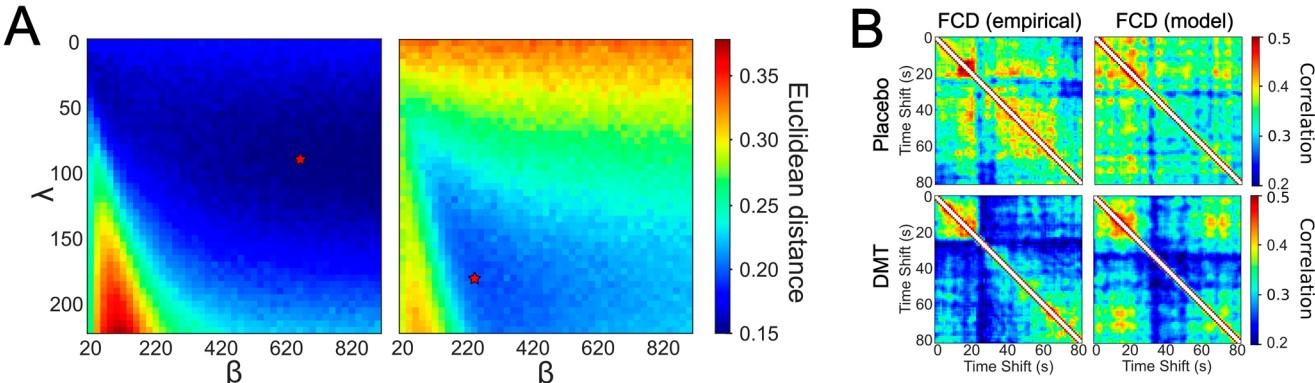

**Fig. 2 | Exploration of parameter space and Functional Connectivity Dynamics (FCD) corresponding to the optimal model parameters. A** Normalized Euclidean distance between simulated and empirical FCD averaged across $n = 50$ simulations for every pair of parameters $\lambda$ and $\beta$. The matrices reveal different peak amplitude ($\lambda$) and latency ($\beta$) values for placebo vs. DMT. Optimal performance for DMT is restricted to a narrower region. The red stars indicate the optimal pair of parameters selected for each condition. **B** Empirical and optimal simulated FCD (columns) for the placebo and DMT conditions (rows) averaged over $n = 15$ subjects (independent simulations). Simulated FCD matrices were computed using optimal $\lambda$ and $\beta$ parameters marked with the red stars in the left panel. Euclidean distances between simulated and empirical FCDs were $0.19 \pm 0.03$ and $0.14 \pm 0.02$ (95% confidence interval [CI]) for DMT and placebo respectively (FCD: functional connectivity dynamics; DMT: dimethyltryptamine).

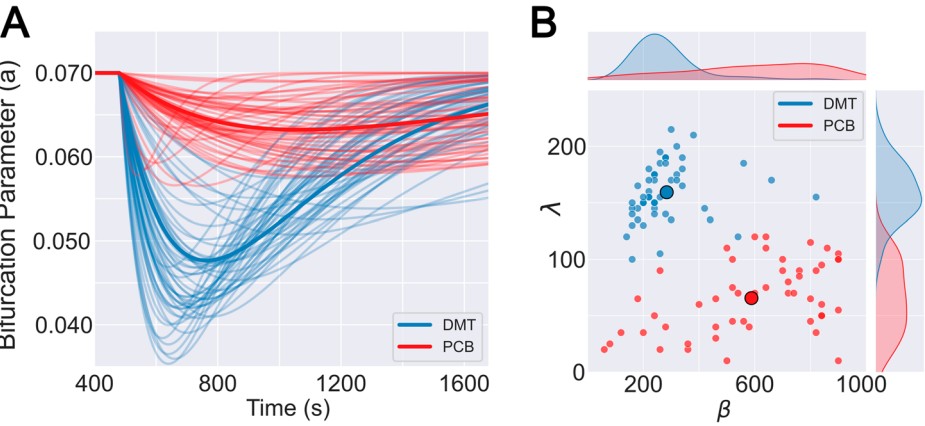

**Fig. 3 | The temporal evolution of the bifurcation parameter, *a(t)*, distinguishes between DMT and the placebo condition. A** $a(t)$, defined as a gamma function, for $n = 50$ independent runs of parameter optimization, compared between both conditions. Plots with thicker lines indicate the curves associated with mean value parameters. **B** Two-dimensional representation of the optimal gamma function parameters, $\lambda$ and $\beta$, both for placebo and DMT. Individual points indicate the outcome of the $n = 50$ independent runs of parameter optimization, while the larger full circles represent the average across simulation runs. Comparisons for the means of both parameters were made with two-sided t-tests with *p*-values < 0.0001 for both parameters (DMT: dimethyltryptamine).

bootstrap procedure which resulted in the distribution of correlation coefficients ($\rho$) presented in the inset of this panel, with a mean value of $\rho = 0.9059 \pm 0.0003$, 95% CI. Figure 5D displays the mean $\rho$ ($\Delta\chi_{max}$ vs. 5HT2a receptor density) across a range of external perturbation intensities ($F_{ext}$). This plot indicates that low stimulation amplitudes exert an effect that is comparatively independent of the local 5HT2a receptor density. However, as the perturbation intensity increases, the receptor density becomes more relevant, peaking at $\rho > 0.9$. Finally, as the intensity keeps increasing, the reactivity decouples again from the receptor density.

## Discussion

This study represents a first step towards the computational modeling of time-dependent psychedelic effects. Our main finding is that DMT destabilizes (i.e. brings closer to the global bifurcation point, or dynamic criticality) whole-brain dynamics, and that the extent of this destabilization is compatible with the characteristic pharmacokinetics of the drug, here constrained by a gamma function[35]. Conversely, an inactive placebo resulted in extreme parameter values that flattened this function, thus approximating a constant value over time. A consequence of this loss of stability is the heightened sensitivity to external perturbations[39], paralleling the dynamics of the bifurcation parameter and thus being maximal when the perturbation is applied in nodes belonging to RSN with high 5HT2a receptor density. This increased sensitivity may affect how the brain responds to incoming sensory stimuli under the effects of the drug, and may also impact on its capacity to amplify endogenous events linked to ongoing mentation and cognitive processing. Note our emphasis on *endogenous events* in this regard, as a converse reduced sensitivity to perturbation may be true in relation to certain external perturbations, such as sensory-evoked potentials[42]. Indeed, recent work showed that external stimulation delivered during the psychedelic state has stronger effects relative to a control condition, which include changes in the intensity of the experience as well as a substantial modification of entropy-based metrics of neural activity[43]. It is also important to note that in the study by Mediano et al. the content and structure of the stimulation was linked to the change in brain entropy, and that the results were partially driven by increased entropy of the baseline condition. In contrast, in our analyses the structured and periodic nature of the external stimulation exerted a comparatively weak effect on the placebo vs. drug condition.

Previous studies have used similar models to investigate the mechanisms underlying psychedelic-induced changes in fMRI resting state activity[26]. However, these studies were concerned primarily with the steady-state effects of the drugs, thus neglecting the analysis of periods when their intensity changes over time, such as the transitions between the baseline and the acute state. These transitions are difficult to capture for the oral administration of LSD or psilocybin, which results in a slower onset and a less marked transition towards the psychedelic state[44,45]. In contrast, the

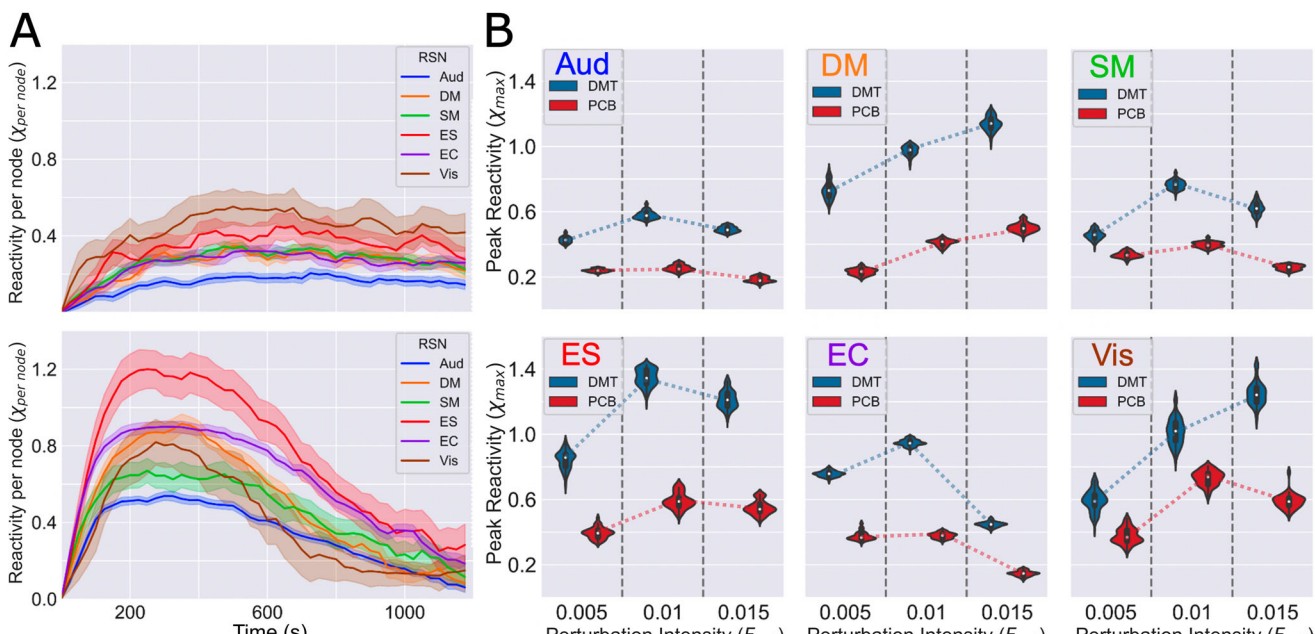

**Fig. 4 | Time-dependent effect of simulated perturbations indicate higher reactivity for DMT vs. placebo. A** Reactivity $\chi(t)$ normalized by the number of nodes in the corresponding RSN, for placebo (top) and DMT (bottom). DMT curves peak around 4 minutes after the dose, restoring baseline values at the end of the session. Placebo shows a lower peak amplitude and longer latencies remaining constant during the whole study. Shaded regions of each line denote the standard deviation of the reactivities ($n = 25$ simulations). **B** Plots of the peak $\chi(t)$ across time ($\chi_{max}$) for each RSN and three different external perturbation intensities ($F_{ext}$) ($n = 200$ bootstrapped samples). Comparisons between conditions (for a given perturbation intensity) and between successive perturbation intensities (for a given condition) were made with two-sided t-tests, giving $p$-values $< 0.0001$ for all comparisons. Executive control (EC) network was an exception when comparing $F_{ext} = 0.005$ and $F_{ext} = 0.01$ ($p = 0.23$) (DMT: dimethyltryptamine; RSN: resting state network).

intravenous administration of DMT results in a fast-onset and relatively brief psychedelic episode that peaks in subjective intensity only a few minutes after the infusion[32] as indeed occurred in our previous work[9,17]. Our current study shows that the whole-brain FCD dynamics induced by DMT on fMRI data recapitulate this temporal evolution, which was not observed for the placebo. It is important to note that even if the intensity of the DMT effects is not reflected on the amplitude of the time-domain fMRI signals, whole-brain FCD contains sufficient information to determine the temporal evolution of the DMT experience as shown by the optimal fit of the model to the FCD data.

The transient destabilization of brain dynamics induced by DMT is consistent with multiple experimental results and theoretical accounts of psychedelic action in the human brain[24]. Drawing from the theory of bifurcations in dynamical systems, as the model approximates the bifurcation point, both the complexity and entropy of simulated brain activity are expected to increase[13–19], together with an expansion in the repertoire of possible metastable states[19]. Also, near the bifurcation point, the sensitivity of the system to external perturbations is maximized[26,39], predicting a larger response to a perturbation within the psychedelic state. A prolonged period of recovery from perturbation is known as *critical slowing* and is considered to be one of several signatures of critical states, such as a global dynamic bifurcation[46]. Using non-invasive transcranial magnetic stimulation (TMS) combined with electroencephalography (EEG), Ort and colleagues could not find changes in cortical reactivity induced by psilocybin; however, they informed changes in spectral content and subjective experience linked to the stimulation[47]. This result is partially consistent with our prediction, especially when taking into account the differences between TMS and a periodic perturbation delivered at the natural frequency of the endogenous oscillations. Empirically, this perturbation could be achieved by non-invasive methods such as transcranial alternating current stimulation (tACS)[48]; however, to date, this form of stimulation has not been investigated in participants undergoing the effects of psychedelic drugs.

Our results draw a parallel between the effects of DMT on whole-brain dynamics and the theory of dynamical and statistical criticality. The main difference between conditions was the behavior of the bifurcation parameter across time, resulting in values closer to the critical value coincident with the peak effects of DMT. This finding, which is consistent with prior work implying increased signatures of criticality in the psychedelic state[46,49,50], can be interpreted through the critical brain hypothesis, stating that the major features of brain dynamics can be explained by proximity to a second-order phase transition[51,52]. Near this critical point, the system exhibits divergent susceptibility (i.e. reactivity), allowing minor perturbations to propagate throughout the entire system[53]. Extending this analogy, we can postulate that 5HT2a receptor activation shifts the network state towards dynamic criticality, which is consistent with multiple other findings related to the brain dynamics under the effects of psychedelic drugs, as discussed by Girn, M. et al 2023[25]. We are mindful, however, that certain specific stimuli consonant with the ongoing state, such as music[54], may have a bigger psychological and, presumably, neurobiological[55] impact under psychedelics. This matter may point to the special value of utilizing naturalistic stimuli with good ecological validity in the context of psychedelic research.

We also report that the sensitivity to external perturbations covariates with the density of 5HT2a receptors. Since we normalized this value by the total number of nodes in each RSN, the explanation for this correlation should be found in the influence of 5HT2a in the organization of functional and structural connectivity. This serotonin receptor subtype is implicated in large-scale heterogeneities of the human cerebral cortex, constituting an important factor to distinguish between unimodal and the higher-level integrative transmodal cortex[56]. Moreover, by bringing the global dynamics closer to dynamical criticality, 5HT2a receptor activation may facilitate the switching between inter-areal coupling that may underly psychedelic-induced neural[57,58] and possibly cognitive flexibility[28], and also open a transient window where the increased functional diversity may exert an effect in the underlying structural connectivity via plasticity effects[59]. Eventually, this could contribute to consolidating the long-term effects

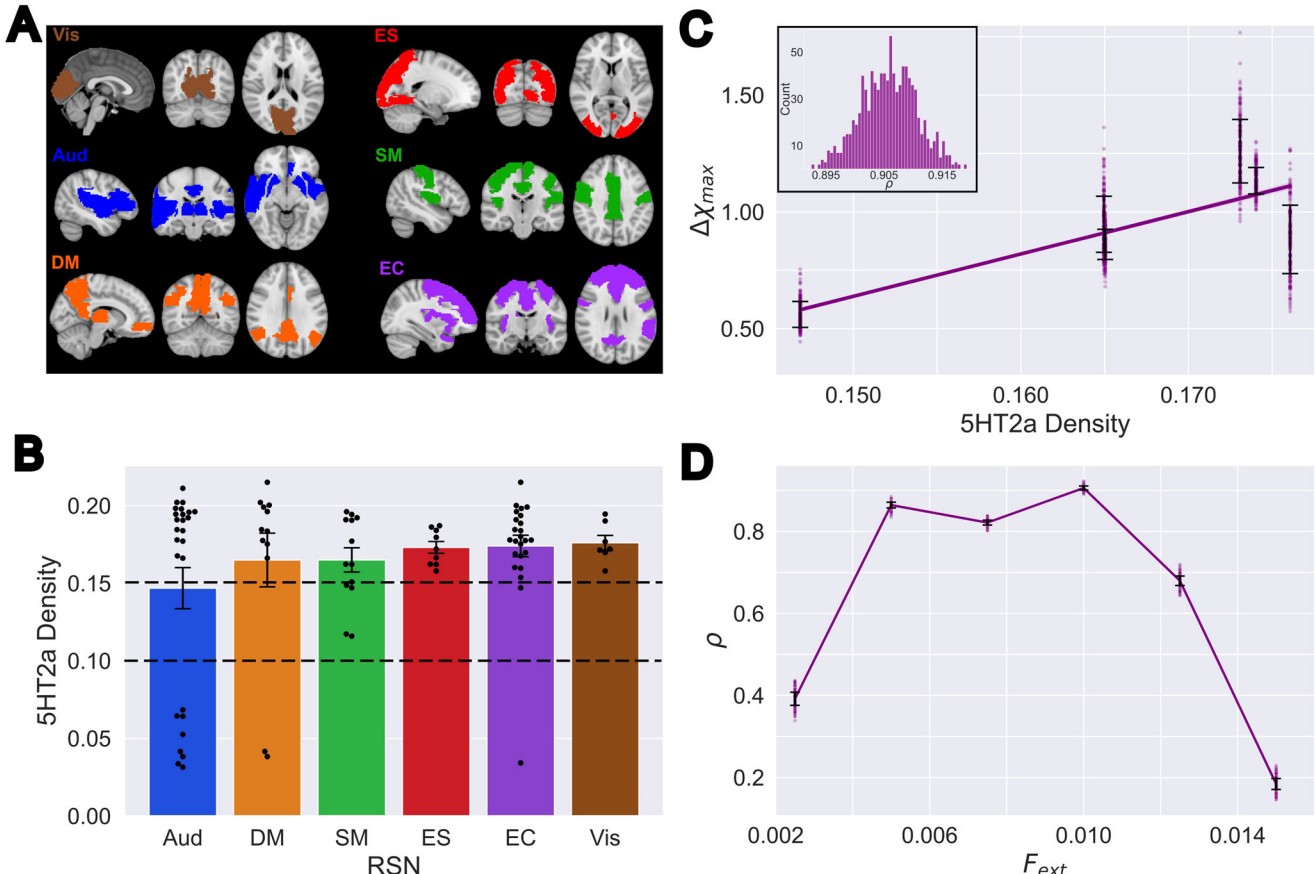

**Fig. 5 | The peak differential reactivity to external perturbations ($\Delta\chi max$) between DMT and placebo correlates with the local 5HT2a receptor density. A** Spatial configuration of the six RSN that constrained the simulated delivery of the perturbation (Vis: primary visual, $n_{Vis} = 7$; ES: extrastriate, $n_{ES} = 9$; Aud: auditory, $n_{Aud} = 26$; SM: sensorimotor, $n_{SM} = 13$; DM: default mode, $n_{DM} = 12$; EC: executive control, $n_{EC} = 24$), adapted from Ipiña, I. P. et al.[40]. **B** 5HT2a receptor density per RSN (individual points and mean ± SE). **C** Peak of $\Delta\chi(t)$ ($n = 200$), $\Delta\chi_{max}$, vs. the 5HT2a receptor density of each RSN together with the best least-squares linear fit. The inset shows the distribution of correlation coefficients ($\rho$) obtained using a bootstrap procedure ($n_{corr} = 1000$). **D** Correlation coefficient $\rho$ (mean ± SD) vs. the external perturbation intensity, $F_{ext}$, $n = 1000$. (DMT: dimethyltryptamine; RSN: resting state network; 5HT: 5-hydroxytryptamine).

associated with brief psychedelic episodes[60]. For high values of the external perturbation amplitude, the reactivity decoupled from the 5HT2a receptor density, likely indicating the saturation of the induced effects on whole-brain dynamics.

We opted to simulate perturbations of the ongoing dynamics with a periodic signal matching the frequency of the local dynamics, i.e. the resonant frequency. This choice represents transcranial alternating current stimulation (tACS) at the peak frequency of the endogenous oscillations. The resonant frequency is a natural choice for stimulation, given that it is known to elicit a maximal response[39]. Also, more complicated signals can be represented in the frequency space via their Fourier decomposition, where the response will be predominantly elicited by the amplitude of the Fourier component at the resonant frequency. Previous studies modeled other forms of stimulation resembling a single transcranial magnetic stimulation (TMS) pulse[38,61,62]. In our model, this would correspond to displacing the dynamics away from the limit cycle and then determining the necessary time to recover baseline dynamics. This choice presents the advantage of representing the effects of a TMS pulse localized in time, one of the most commonly used forms of transcranial stimulation. However, it is unlocalized in the frequency domain, and therefore it does not inform how the model responds to specific frequencies.

Beyond the current application, the temporal parametrization of whole-brain models may find other uses to test whether a certain temporal process underlies the recorded neuroimaging data, and to inform the potential mechanisms behind the empirical observations. The signal pre-processing steps applied to resting state fMRI data typically include high

pass filters which are necessary to attenuate the effects of scanner drift and head movement[63]. Therefore, the fMRI time series may not directly reflect the dynamics of the phenomenon under study, unless the experiment is designed to maximize the signal-to-noise ratio, e.g. with tasks or stimulation structured in according to an adequate block design[64]. However, it may not be possible to control the onset and length of the temporal process under study, as in the current application to the study of a transient pharmacologically induced event. This could also occur in the study of seizures[65], sleep-related transients and graphoelements (e.g. spindles, K-complexes)[66], spontaneous mentation or ongoing cognition[67], and in other endogenous phenomena with an interesting temporal structure beyond experimental control. In these cases, the use of time-dependent whole-brain models may be useful to detect the fingerprints of the event in the FCD, by comparing different temporal parametrizations by fitting and contrasting their corresponding goodness of fit.

One important limitation of our approach is that the temporal parametrization a priori constrains the model and its capacity to detect the underlying temporal dynamics. By choosing a gamma function, our approach only allowed us to test whether the FCD dynamics under DMT could be better explained by this specific temporal evolution of the bifurcation parameter relative to the placebo condition. However, since we did not specify a prior direction of the effect, this approach allowed us to test whether DMT could bring the dynamics closer to the global bifurcation and therefore towards a point of decreased stability. When comparing the gamma function with a null model keeping $a(t)$ constant post-dose, we found that this parametrization outperformed the gamma function for the

placebo condition, as expected from our results. However, we did not find significant differences relative to the gamma function for the DMT condition (see supplementary Fig. 3). We note that within the family of gamma functions used to parametrize the bifurcation parameter, the optimal fit was consistently obtained in a region of parameter space corresponding to a relatively early peak at 10 min, and a peak value that was in all cases larger than the peak value obtained for the placebo condition. These parameter values are compatible with human studies of DMT pharmacokinetics[68]. The lack of difference observed between both models could depend on methodological choices (e.g. the choice of the optimization metric and the need to balance relative differences between matrix values vs. the overall magnitude of the matrix entries), which could be addressed in future studies fitting whole-brain models to transient brain states.

The choice of the gamma function can be justified by the prior literature on the time course of DMT-induced effects, as well as its effects on EEG activity and their correlation with the drug pharmacokinetics[9,16,17,32]. While more complex models could be implemented to adequately describe these dynamics[33], our choice has the merit of constituting a qualitative description of the transient effects of DMT without incurring in a proliferation of free parameters required for more neurobiological realism. Moreover, the comparatively poor temporal resolution of fMRI may limit the potential gain of including a more nuanced description of pharmacodynamics. This limitation could be addressed by reproducing our work with data from faster imaging modalities (such as EEG and MEG). Another limitation is the number of participants ($n = 15$), which is comparatively low for a pharmacological fMRI study. In the case of this study, after applying strict criteria to exclude subjects due to head motion in the scanner only 75% of the original data was included. While we modeled group-averaged FCD matrices, future studies at the single-subject level should attempt to raise the effective number of subjects included in the model. Finally, psychedelics are known to introduce physiological effects (e.g. heart rate, blood pressure, respiration), as well as alterations to the coupling between neural activity and the cardiovascular system[69]. In our study, we followed previous research by modeling physiological noise in terms of the average signal measured at the ventricles, draining veins, and local white matter. While this is a standard procedure in the field[8,70–73], there could be limitations associated with this approach which could be overcome by introducing regressors based on cardiac and respiratory time series[74]. The limitations imposed by changes to neurovascular coupling could be addressed in future studies by using a condition-specific hemodynamic response, estimated either from task and/or stimulation paradigms[75], or by deconvolution of resting state BOLD signals[76].

In summary, our work shifts the focus from the reproduction of the steady-state effects of psychedelics towards disentangling the dynamics of the fast-onset and short-lived intravenous DMT experience. The future implementation of more realistic biophysical models could contribute to our understanding of how the interaction between drugs, neurotransmitters and receptors is capable of initiating a cascade of neural events which ultimately results in a global brain state associated with profound alterations in consciousness and cognitive processing with potentially lasting consequences[60]. The combination of these models with a more nuanced description of drug pharmacokinetics and pharmacodynamics could also contribute to explain the potential multiphasic effects of some psychedelic compounds, and to test potential mechanisms behind their long-lasting effects, which are a key aspect of their possible therapeutic use in patients with depression and other psychiatric disorders.

## Methods
### Study participants and experimental design
This is a re-analysis of a previously published EEG-fMRI dataset acquired from healthy participants under the acute effects of DMT[9]. The original publication can be referenced for in-depth methodological details.

The study followed a single-blind, placebo-controlled design, with all participants providing written informed consent. The experimental protocol received approval from the National Research Ethics Committee

London—Brent and the Health Research Authority, conducted in adherence to the Declaration of Helsinki (2000), the International Committee on Harmonization Good Clinical Practices guidelines, and the National Health Service Research Governance Framework. All ethical regulations relevant to human research participants were followed.

Volunteers complete two visits at the Imperial College Clinical Imaging Facility, spaced two weeks apart. On each testing day, participants were subject to separate scanning sessions. In the initial session, they received intravenous administration of either placebo (10 mL sterile saline) or 20 mg DMT (fumarate form dissolved in 10 mL sterile saline). This was done in a counter-balanced order, with half receiving placebo and the rest receiving DMT. The first session comprised continuous resting-state scans lasting 28 minutes, with DMT/placebo administered at the end of the 8th minute, and scanning concluding 20 minutes after injection. Participants lay in the scanner with closed eyes aided by an eye mask, while fMRI data was recorded. In the second session, participants were cued to verbally rate the subjective intensity of drug effects every minute in real-time. Only fMRI data from the first scanning session was used for the present analysis.

A cohort of 20 participants completed all study visits, consisting of 7 females and a mean age of 33.5 years and a standard deviation of 7.9. For the present study data from only 15 of the subjects were used, the rest being discarded due to strong head movement artifacts inside the scanner (see below for further information on the exclusion criterion). This final sample consisted of a cohort with 5 females and a mean age of 39.6 and a standard deviation of 9.6.

### fMRI acquisition and preprocessing
Images were acquired using a 3 T MR scanner (Siemens Magnetom Verio syngo MR B17) with a 12-channel head coil for compatibility with EEG acquisition. Functional imaging was performed using a T2*-weighted BOLD-sensitive gradient echo planar imaging sequence [repetition time (TR) = 2000 ms, echo time (TE) = 30 ms, acquisition time (TA) = 28.06 min, flip angle (FA) = 80°, voxel size = $3.0 \times 3.0 \times 3.0$ mm$^3$, 35 slices, interslice distance = 0 mm].

The preprocessing steps involved despiking, slice time correction, motion correction, brain extraction, rigid body registration to anatomical scans, nonlinear registration to 2 mm MNI brain, denoising (via Independent Component Analysis), and scrubbing (using a framewise displacement [FD] threshold of 0.4, with scrubbed volumes replaced by the mean of surrounding volumes)[77]. The BOLD time series were filtered with a second-order Butterworth filter in the range between 0.01 and 0.08 Hz, in line with previous studies[78–81]. The choice of the low-pass cutoff was justified as it helps to filter out physiological noise, which tends to dominate higher frequencies[78]. Further steps included spatial smoothing (FWHM) of 6 mm, linear and quadratic detrending, and regressing out nine nuisance regressors. These regressors, all band-pass filtered, consisted of six motion-related parameters (3 translations, 3 rotations) and three anatomically related parameters (ventricles, draining veins, local white matter).

Finally, time series were extracted for each Automated Anatomical Labeling (AAL)[82] template region by averaging the time series from all voxels within the corresponding region.

Out of 20 participants, five were excluded from group analyses due to excessive head movement during the 8-minute post-DMT period ( > 20% of scrubbed volumes with a framewise displacement threshold [FD] of 0.4).

### Anatomical connectivity matrix (SC)
The structural connectome was obtained by applying diffusion tensor imaging (DTI) to diffusion-weighted imaging (DWI) recordings from 16 healthy right-handed participants (11 men and 5 women, mean age: 24.75 ± 2.54 years) recruited online at Aarhus University, Denmark. Subjects with psychiatric or neurological disorders (or a history thereof) were excluded from participation. The MRI data (structural MRI, DTI) were recorded in a single session on a 3 TS Skyra scanner. The following parameters were used for the structural MRI T1 scan: voxel size of 1 mm$^3$; reconstructed matrix size $256 \times 256$; echo time (TE) of 3.8 ms and repetition time (TR) of 2300 ms.

DWI data were collected using the following parameters: TR = 9000 ms, TE = 84 ms, flip angle = 90°, reconstructed matrix size of 106 × 106, voxel size of 1.98 × 1.98 mm with slice thickness of 2 mm and a bandwidth of 1745 Hz/Px. Furthermore, the data were recorded with 62 optimal nonlinear diffusion gradient directions at b = 1500 s/mm². Approximately one non-diffusion-weighted image (b = 0) per 10 diffusion-weighted images was acquired. Additionally, the DTI images were recorded with different phase encoding directions. One set was collected applying anterior to posterior phase encoding direction and the second one was acquired in the opposite direction. The AAL template was used to parcellate the entire brain into 90 regions (76 cortical regions and 14 subcortical regions). The parcellation contained 45 regions in each hemisphere. To co-register the EPI image to the T1-weighted structural image, the linear registration tool from the FSL toolbox (www.fmrib.ox.ac.uk/fsl, FMRIB, Oxford)[83] was employed. The T1-weighted images were co-registered to the T1 template of ICBM 152 in MNI space. The resulting transformations were concatenated, inverted and further applied to warp the AAL template from MNI space to the EPI native space, where the discrete labeling values were preserved by applying nearest-neighbor interpolation. SC networks were constructed following a three-step process. First, the regions of the whole-brain network were defined using the AAL template. Second, the connections between nodes in the whole-brain network (i.e., edges) were estimated using probabilistic tractography for each participant. Third, results were averaged across participants.

Data preprocessing was performed using FSL diffusion toolbox (Fdt) with default parameters. Following this preprocessing, the local probability distributions of fiber directions were estimated at each voxel[84]. The probtrackx tool in Fdt was used to provide an automatic estimation of crossing fibers within each voxel, which has been shown to significantly improve the tracking sensitivity of non-dominant fiber populations in the human brain[85]. The connectivity probability from a seed voxel $i$ to another voxel $j$ was defined by the proportion of fibers passing through voxel $i$ that reached voxel $j$ (sampling of 5000 streamlines per voxel). All the voxels in each AAL parcel were seeded (i.e. gray and white matter voxels were considered). This was extended from the voxel level to the region level, i.e. in a parcel consisting of $n$ voxels, $5000 \times n$ fibers were sampled. The connectivity probability $P_{ij}$ from region $i$ to region $j$ was calculated as the number of sampled fibers in region $i$ that connected the two regions, divided by $5000 \times n$, where n represents the number of voxels in region $i$. The resulting SC matrices were thresholded at 0.1% (i.e. a minimum of five streamlines).

Due to the dependence of tractography on the seeding location, the probability from $i$ to $j$ was not necessarily equivalent to that from $j$ to $i$. However, these two probabilities were highly correlated across the brain for all participants ($r > 0.70$, $p < 10 - 50$). As the directionality of connections cannot be determined using diffusion MRI, the unidirectional connectivity probability $P_{ij}$ between regions $i$ and $j$ was defined by averaging these two connectivity probabilities. This unidirectional connectivity was considered a measure of SC between the two areas, with $C_{ij} = C_{ji}$. The regional connectivity probability was calculated using in-house Perl scripts. For both phase encoding directions, 90 × 90 symmetric weighted networks were constructed based on the AAL parcellation, and normalized by the number of voxels in each AAL region, thus representing the SC network organization of the brain of each participant. Finally, the data was averaged across participants. (Supplementary Fig. 4 summarizes the similarity between structural and functional connectivity, both for simulated and empirical data).

## Whole-brain computational model

We simulated brain activity measured with fMRI at the whole-brain level by using a Stuart–Landau oscillator (i.e. Hopf normal form) for the local dynamics (see Fig. 1, "Local model"). This phenomenological model aims to directly simulate recorded brain signals. The emergent global dynamics are simulated by including mutual interactions between brain areas according to the anatomical connectivity matrix $C_{ij}$ obtained from DTI (see Fig. 1, "Inter-areal coupling"). The full model consists of 90 coupled nodes representing the 90 cortical and subcortical brain areas from AAL parcellation, with the following temporal evolution for region $n$:

$$\frac{dx_n}{dt} = \left(a(t) - x_n{}^2 - y_n{}^2\right)x_n - \omega y_n + G\sum_{p=1}^{90} C_{np}(x_p - x_n) + \gamma \eta_n(t)$$

$$\frac{dy_n}{dt} = \left(a(t) - x_n{}^2 - y_n{}^2\right)y_n + \omega_n x_n + G\sum_{p=1}^{90} C_{np}(y_p - y_n) + \gamma \eta_n(t)$$

(1)

Here, $\eta_n$ represents additive Gaussian noise with standard deviation γ (set to 0.05), $C_{np}$ are the matrix elements of the anatomical connectivity matrix, $G$ is a factor that scales the anatomical connectivity (fixed at $G = 0.5$, as determined previously elsewhere[40]; see also section "Model fitting to baseline data"), $\omega_n$ is the peak frequency of node $n$ in the 0.01-0.08 Hz band (determined using Fourier analysis applied to the empirical time series and averaged across participants, see Supplementary Fig. 5 for the distribution of node frequencies), and variable $x_n$ represents the empirical brain activity signal of node $n$, which was used to compute the simulated FC and FCD matrices. The simulated time series were not band-pass filtered, given that the frequency of each node was defined based on the empirical data, and that the positive optimal bifurcation parameters resulted in oscillating behavior with frequencies within the range of the empirical values.

The local dynamics present a supercritical bifurcation at $a = 0$, so that if $a > 0$ the system engages in a stable limit cycle (i.e. oscillations) with angular frequency $\omega_n$, and when $a < 0$ the local dynamics are attracted to a stable fixed point representing a low activity state dominated by noise. Close to the bifurcation, the additive noise can induce a switching behavior between regimes, resulting oscillations with a complex envelope[36].

To model the short-lasting effects of DMT, we introduce a time-dependent bifurcation parameter $a(t)$ following a gamma function that represents a simplified description of drug pharmacokinetics[35]:

$$a(t) = \lambda\left(\frac{te^{-t/\beta}}{N}\right)$$

(2)

where $N$ is a constant that normalizes the term between brackets, $\lambda$ is a scale parameter determining the amplitude of the peak, and $\beta$ the parameter that controls the rate of decay of the function, and therefore is related to the latency of the peak (see Fig. 1, "Time dependency").

All codes necessary for the implementation of the model are available online at https://github.com/juanpiccinini/DMT-whole-brain.

## Model fitting to empirical data

To characterize the time-dependent structure of brain dynamics, we computed the FCD matrices (see Fig. 1, "Functional connectivity dynamics")[37]. For this purpose, each full-length signal of 28 min was split up into $M = 82$ sliding windows of 60 s each, with an overlap of 40 s between successive windows. For each sliding window centered at time t, the functional connectivity matrix FC(t) was computed. The FCD matrix is a $M \times M$ symmetric matrix whose $t_1$, $t_2$ entry is defined by the Pearson correlation coefficient between the upper triangular parts of the two matrices FC($t_1$) and FC($t_2$). These matrices were computed for each of the fifteen participants and simulations by exhaustively exploring the model parameters related to the temporal evolution of the bifurcation parameter, $\lambda$ and $\beta$. To compare the FCD matrices taking into account their temporal structure, we used the Euclidean distance between the elements of the empirical and simulated matrices. In Supplementary Fig. 6, we compare the obtained FCD matrices with an alternative construction based on phase coherence, while the agreement between sliding windows and phase coherence is summarized in Supplementary Fig. 7. An alternative analysis based on metastability and synchrony is presented in Supplementary Fig. 8.

## Model fitting to baseline data

To fit the model to the baseline data before drug administration, we first conducted a search for the optimal values of the bifurcation parameter $a$ and the coupling parameter $G$. This was done by fitting the FCD submatrix comprising the 22 temporal windows (corresponding to the first 8 minutes before DMT injection) averaged over all subjects during both sessions (30 submatrices in total). We performed an exhaustive parameter space exploration by varying the two free parameters of the model: $a$ was changed homogeneously across all nodes from –0.1 to 0.1 in increments of 0.01, while $G$ was varied from 0 to 2 in steps of 0.1. This procedure was performed 30 times for each pair of parameters, and the resulting distance metrics were then averaged to determine the optimal parameters (see Supplementary Fig. 9A). We observed that the optimal values spanned an extended region in parameter space, which included $G = 0.5$, a value consistent with previous determinations of the coupling parameter in similar datasets[40]. While the global minimum was found for another value of $G$, the resulting goodness of fit was almost identical to the value obtained for $G = 0.5$, without significant differences between both choices (see panel B of Supplementary Fig. 9). With the choice of $G = 0.5$ and $a = 0.07$, the Euclidean distance found for the baseline was $0.084 \pm 0.005$, 95% confidence interval [CI]. Supplementary Fig. 10 presents a complementary comparison of the normalized Euclidean distance and the Pearson correlation computed between the empirical and simulated FCD and FC matrices.

## Fitting the temporal evolution of the bifurcation parameter

We fixed the value of the bifurcation parameter to the baseline value of $a = 0.07$ for the first 8 minutes of the simulation and introduced the time dependency afterwards, given by the difference between the baseline value and the gamma function $\lambda \left( \frac{t e^{-t/\beta}}{N} \right)$, with $t = 0$ here indicating the time of the drug injection. Next, we performed an exhaustive exploration of the parameter space spanned by $\lambda$ and $\beta$, searching for the optimal combination of parameters. For this purpose, we explored a grid given by $\lambda = [0, 200]$ and $\beta = [20, 900]$ in steps of 5 and 20 units, respectively. For each parameter combination, we computed the FCD 15 times (i.e. once per participant) randomly changing the initial conditions of the model. The resulting FCD matrices were averaged and compared with the empirical FCD using the Euclidean distance. To compare the FCD matrices taking into account their temporal structure, we used the Euclidean distance between the elements of the empirical and simulated matrices, and then we normalized the results dividing by the empirical norm to account for the relative change. The procedure described in this section was repeated 50 times for each pair of parameters.

## Simulated perturbations

Next, we modeled an oscillatory perturbation and investigated the response of the whole-brain model fitted as explained in the previous section. The perturbation was given by an external additive periodic forcing term applied to node $n$, $\mathbf{F_n} = \mathbf{F}_{ext}(cos(\omega_n t) + \mathbf{i}\, sin(\omega_n t))$, where the real part of the perturbation is added to the equation for $\frac{dx_n}{dt}$ in Eq. 1 and the imaginary part to the equation for $\frac{dy_n}{dt}$. In the equation for $\mathbf{F_n}$, $\omega_n$ is the natural frequency of the time series corresponding to node $\mathbf{n}$ (same as in Eq. 1).

 To facilitate the interpretation of the results, we applied this perturbation to nodes located within six different resting state networks (RSN) identified using independent component analysis (ICA) by Beckmann, et al.[41]. To account for the time variation of the reactivity, we sampled equally spaced values of $a(t)$, here noted $a_p$, with $p$ indexing the time sample. In total we ended up with 42 $a_p$ values corresponding to the gamma function of each condition sampled at these time points. Then, for every one of those values, we performed a simulation keeping $a_p$ constant until the end of the simulation, i.e. for every simulation we set $a = 0.07$ for the first 8 minutes, corresponding to the pre-dose time interval, and then keeping the constant value $a_p$ until the end of the simulation. Therefore, the functional form of the bifurcation parameter is given by the concatenation of two constant functions, $a = 0.07$ and $a_p$, with $p = 0,…,42$. This procedure allowed to compute how the dynamics responded to the external perturbation at each $a_p$ value over the extended period of time that was used to obtain the FCD. Regarding the stimulation, we applied the perturbation after the first 8 minutes corresponding to the baseline, varying the amplitude $F_{ext}$ from 0 to 0.015 in 0.0125 steps. The maximum value of this range was chosen as higher values saturated the local reactivity, flattening the curves. To summarize, for each combination of RSN, amplitude $F_{ext}$ and the value of $a_p$, we computed the resulting FCD and its distance to the empirical condition, and then assessed the impact of the perturbation as explained in the following section.

## Measure of reactivity to perturbations

We interpret the whole-brain model reactivity as the sensitivity of brain activity to changes in the external periodic stimulation. Following an analogy with the concept of susceptibility in statistical physics, we defined the reactivity as the following derivative:

$$\chi(t) = \frac{\partial M}{\partial F_{ext}} \tag{3}$$

where $M$ denotes the Euclidean distance between the simulated and empirical FCD matrices. As $F_{ext}$ is increased, we expect the stimulated FCD to depart from the baseline empirical value. $\chi(t)$ measures the rate at which this divergence occurs. Thus, a large $\chi(t)$ value indicates that at time $t$, the effect of changing $F_{ext}$ is maximal, measured in terms of its impact on the Euclidean distance between the simulated and empirical FCD matrices. Conversely, a small $\chi(t)$ represents a regime were changing $F_{ext}$ exerts comparatively little impact on the FCD. The reactivity $\chi(t)$ was computed using a second order finite difference method. We evaluated $\chi(t)$ relative to its value at $t = 0$ by subtracting $\chi(t = 0)$ at later times. This was done in order to capture the changes of the perturbation relative to the baseline part. Furthermore, given that the number of nodes differs across the RSNs, and that the reactivity can depend on the number of stimulated nodes, we normalized its value by the number of nodes of each RSN.

## Bootstrap method

To determine the peak of $\chi(t)$ across the duration of the simulation, we used a bootstrapping procedure to obtain a distribution of peak values which allowed us to compute a confidence interval for the resulting average value. The bootstrap procedures were conducted by randomly drawing samples (with replacement) from the distribution of values, creating different $\chi(t)$ curves in each iteration and measuring its maximum value. The size of the sampled subset was equal to that of the original distribution. This procedure was repeated 200 times, generating a bootstrap distribution of the desired magnitude. When calculating the correlations between local 5HT2a receptor density and the maximum reactivity per RSN, the bootstrap was done 1000 times to generate the histograms.

## Statistics and Reproducibility

95% Confidence Intervals were calculated using the standard error of the mean (SEM) and a z-score of 1.96. Metric comparisons were achieved using a two-sided paired t-test assuming non-equal variance between pair of metrics. To determine the peak location and significance of $\chi(t)$ across the duration of the simulation, we implemented a bootstrap procedure (see section "*Bootstrap method*"). A Kolmogorov Smirnov test was used in Supplementary Fig. 3. Effect size to compare model simulations in Supplementary Fig. 9 were calculated using Cohen's d.

## Receptor density maps

The receptor density maps used were estimated using PET tracer studies obtained by Hansen and colleagues[86]. All PET images were registered to the MNI-ICBM 152 nonlinear 2009 (version c, asymmetric) template and subsequently parcellated to the 90 region AAL atlas[82]. For more details on the acquisitions and limitations of the data set see the original publication[86].

**Article**

## Reporting summary

Further information on research design is available in the Nature Portfolio Reporting Summary linked to this article.

## Data availability

Source data to reproduce figures is available on https://github.com/juanpiccinini/DMT-whole-brain.git along with the code[87] (See Code availability). All other data is available from the corresponding author on reasonable request.

## Code availability

Code needed to simulate the model and evaluate the conclusions in this paper are available at https://github.com/juanpiccinini/DMT-whole-brain.git[87].

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

## Acknowledgements

E.T. is supported by PICT-2019–02294 (Agencia I + D + i, Argentina), ANID/FONDECYT Regular 1220995 (Chile) and by PIP 1122021010 0800CO (2022-2024) (CONICET, Argentina). The collection of the empirical data used in this manuscript was funded via a donation by Patrick Vernon, mediated by the Beckley Foundation. C.T. is funded by a donation by Anton Bilton to the Centre for Psychedelic Research.

## Author contributions

J.I.P., Y.S.-P., C.P. and E.T. designed the research. J.I.P. conducted the research. J.I.P., Y.S.-P. and E.T. analyzed and interpreted the results. J.I.P. created the code and made the figures. D.N, R.C.-H. and C.T. provided the curated data. J.I.P. and E.T. wrote the manuscript. C.P., G.D, M.K. provided analytic support. Y.S.-P. and E.T. supervised the research. All authors edited the manuscript.

## Competing interests

The authors declare no competing interest. E.T. is an Editorial Board Member for Communications Biology, but was not involved in the editorial review or the decision to publish this article.
