## [Peer review file · Communications Biology]

Transient destabilization of whole brain dynamics induced by N,N-Dimethyltryptamine (DMT)

Corresponding Author: Mr Juan Ignacio Piccinini

Version 0:

Reviewer comments:

Reviewer #1

(Remarks to the Author)

Reviewer #2

(Remarks to the Author)

The manuscript addresses the dynamics of the resting-state brain induced by psychedelic drugs (DMT). For this, a whole-brain model of coupled limit-cycle oscillators was investigated with time-dependent bifurcation parameter $a(t)$ supposed to mimic the impact of drugs on the brain dynamics.

It was in particular shown that, starting from an oscillatory regime with positive parameter value in the baseline, this parameter decreased towards the critical bifurcation value after the drug administration and eventually recovered to the baseline after a transient of about 30 min. This happened for the placebo condition as well, however, the parameter variability in this case was much lower and latent as compared to the case of the true drug injection. Furthermore, under the DMT condition, the model exhibited a pronounced reaction on external periodic forcing as reflected by a so-called model reactivity calculated in the manuscript. The reactivity difference to the placebo condition was found to correlate with the 5HT_{2a} receptor density for a few considered resting-state networks receiving the perturbations.

This is an interesting study making a valuable contribution to the modeling of the transient brain dynamics under the drug application. The manuscript however requires a revision, where a few critical comments below are suggested to be addressed before any recommendation concerning its publication could be made.

1. After the model was fitted to empirical data (FCD) during parameter optimization, it is supposed to reflect the brain dynamics during the drug administration. I however missed convincing estimations of how close the modeling results replicate the empirical observations.

I appreciate that the FCDs were compared based on the Frobenius distance, which is supposed to better reflect the closeness of FC dynamics, and which would be missed otherwise for a statistical comparison of the FCD distributions. But it is difficult to interpret the obtained distances (Fig.2). Calculating the correlation between the empirical and simulated FCDs may better reflect the pattern similarity and provide an evidence of how similar they are to each other in DMT and placebo cases.

Because of the well-defined reference time point of drug administration, why were the empirical and simulated FC matrices calculated at every sliding window not compared to each other at these time points? Such a comparison could also show how well the model with varied parameter followed the recorded brain activity.

2. Before coming to periodic stimulation of the obtained model with optimized parameters, where the corresponding empirical data are absent as far as I can infer, its output can be compared with other empirical observations available from the studies cited in the manuscript. For example, Ref. [9] demonstrated a few interesting results for the same data used in the current manuscript, where "... fMRI results revealed robust increases in global functional connectivity (GFC), network disintegration and desegregation, and a compression of the principal cortical gradient under DMT."

By confirming the reported observations for the modeling results could enhance the confidence in the model and its other outcomes, where a direct comparison with empirical data is not possible so far, which may question the respective modeling results with respect to their correspondence to reality. Therefore, the results of study [42] should be discussed in more detail, where in particular was shown that stronger stimuli were accompanied by a weaker differentiation between drug-induced and placebo conditions, which might be contrasted with the presented modeling results.

3. Through the entire study the coupling parameter was fixed to $G = 0.5$ and taken from another study [39] dedicated to very different modeling conditions and data. It is not immediately evident why this was justified, especially because this parameter could be important to appropriately fit the model. Furthermore, if the bifurcation parameter $a(t)$ was allowed to vary, why was this denied for the coupling parameter, where they may influence each other and impact the modeling results and conclusions, in particular, "increases in global functional connectivity (GFC)" mentioned above?

4. Similar question arises from the considering the group-averaged structural connectivity (SC) from different data. If we proceed with such assumptions even further to extreme, then we may claim that there exists one SC (and one value of the coupling parameter) that can be considered for modeling whatever (healthy) functional data we take for whatever modeling conditions, which is confusing and difficult to accept.

5. The manuscript also investigated the sensitivity of the model to perturbations, where a model reactivity was calculated. Unfortunately, the methodological approach (lines 516-527) is very unclear and confusing, and should be better explained and justified.

It is not clear what the "equally spaced values of $a(t)$ " are. Then, the simulations appeared to be performed for constant values of the parameter, $a=0.07$ for the pre-dose time interval and for some constant a_p values after the drug administration. We however learned that $a(t)$ should vary with time during this epoch. The transient model dynamics and its (transient) reaction on the perturbations might be different for these two conditions of constant and time-varied a .

"The perturbation was applied after the first 8 minutes corresponding to the baseline, varying the amplitude F_{ext} from 0 to 0.015 in 00125 steps." Can I understand this so that the amplitude was varied (increased) with time instead of parameter a ? If true, why not otherwise?

Next, "The maximum value of the range was chosen" - which value, from which range and chosen for what and why? This was done because "it saturated the correlation between 5HT_{2a} receptor density and local reactivity, as shown in Fig. 5". Here I may see some indications of a circular logic, where the information about correlation was used to obtain results that will be employed to demonstrate the correlations.

In Fig.5 we see a non-monotonic behavior of the correlation versus the stimulation intensity with smaller correlation for large/small forcing. The large values of intensity were chosen, if I read it correctly, but again, for what, especially because all combinations of the parameters were then considered: "For each combination of RSN, F_{ext} and a_p we computed the resulting FCD and its distance to the empirical condition"?

6. The used measure $\chi(t)$ of the sensitivity/reactivity is complicated and difficult

to interpret. Additional explanations are necessary in this respect. I may also suggest that the reported results for sensitivity (Fig. 4) will be complemented by simple illustrations on how the FC matrices were affected over time by different stimulation intensities for DMT and placebo conditions starting from no-stimulation case.

7. As mentioned in Ref. [9], "Values related to head motion (i.e., frame-wise displacement; FD) were significantly different between DMT and placebo", where 13 subjects were included in the dynamical analysis in contrast to 15 subjects in the current study. In that study two more subject sub-samplings were considered to address this problem, which is applicable also to the current manuscript. Can the motion artifacts affect the modeling results, or is the model little sensitive to them?

Reviewer #3

(Remarks to the Author)

In this article, the authors propose a phenomenological model of brain network dynamics in which they model the brain as a dynamic system and parametrize the proximity of the brain to a critical state transition as a time-varying parameter. The goal of this modelling approach is to describe how the brain is moving closer to a critical state under the influence of DMT. They find that the brain approaches a critical state after DMT administration and show in simulations that this places the brain in a more receptive state where external perturbations have a larger effect on the system, the extent of which correlates with 5-HT 2A receptor density, a known pharmacological target of psychedelics.

Overall, this paper is interesting and well-written. I believe that modelling the effects of psychedelics is very valuable to formally test theories of psychedelics like brain entropy accounts and forces us to formalize these theories by making specific modelling choices. This work will be of interest to a broader readership.

I do, however, have a few remaining concerns and suggestions for improvement outlined below. I hope you find my comments helpful and constructive.

Major points:

Methods

- Psychedelics can have drastic effects on physiological effects (heart rate, blood pressure, etc.). Especially when it comes to functional connectivity analyses, these effects can induce spurious correlations between regions that are for example close to the same blood vessel. I noticed that the authors have followed the usual preprocessing steps for FC analyses, but do they have ongoing measures of heart rate and pulse during the fMRI scan which they could use to remove artifacts by explicitly modelling these artefacts (e.g. using this approach: <https://doi.org/10.1016/j.jneumeth.2016.10.019>). I believe that usual FC correction methods might not be good enough to account for these confounding factors. If this information is not available this should at least be clearly stated as a limitation.

- Could the authors provide a justification for using an anatomical parcellation over a functional parcellation (e.g., Schaefer Atlas) considering that they are interested in functional connectivity?

- When it comes to the modelling, I am not convinced that modelling the placebo response with a gamma function makes sense. This choice seems reasonable for the DMT condition, but less well suited for placebo in the absence of a pharmacological intervention. Indeed eyeballing Figure 4, the gamma function in the placebo condition appears to approximate a constant effect. Could the authors model the time-varying effect as a constant term + a gamma function that is only active in the DMT condition? Perhaps the authors could justify this choice, or consider changing their model for the placebo condition. Alternatively, I would like to see a comparison with a null-model with a constant a parameter. Presumably, this model would perform better or as good in the placebo condition, but worse in the LSD condition, which would be informative to show.

- Figure 4: I am not quite sure, what the advantage of estimating the gamma function from fMRI data compared to using the known pharmacogenetics of DMT is. Is this just to demonstrate that it can be estimated from this data? Could the authors clarify what the advantage of estimating it from the data is. Perhaps the authors could also assess the correlation between a canonical pharmacogenetics function and their estimated functions.

- Figure 4B: Reactivity depends on stimulation intensity. Could the authors explain why stronger perturbations lead to less reactivity for the EC network. In the discussion (p.15, ll. 326ff) the authors argue: "For high values of the external perturbation amplitude, the reactivity decoupled from the 5HT2a receptor density, likely indicating the saturation of the induced effects on whole-brain dynamics." This explanation is plausible for networks where the increasing stimulation leads to a plateau in reactivity (maybe Aud, ES and SM), but does this really explain why the reactivity goes down in EC? Perhaps the authors could comment on this.

Version 1:

Reviewer comments:

Reviewer #1

(Remarks to the Author)

Thank you for addressing my comments.

Reviewer #2

(Remarks to the Author)

I have no any further comments, and the revised manuscript can be published in its present form.

Reviewer #3

(Remarks to the Author)

I thank the authors for addressing my comments carefully. I have only one remaining concern.

In my initial review I asked the authors whether a constant $\alpha(t)$ would be the better model for the placebo condition. The authors now included a comparison with this null model in the supplement, which I greatly appreciated. The authors say that the constant model significantly outperforms the gamma model in the placebo condition as I had suspected, but none of the other comparisons are significantly different. I am a bit surprised that they do not find a significant difference in favor of the gamma model in the DMT condition. Doesn't this suggest that a constant model would be as good in explaining the data as the gamma model even in the DMT condition, which would be a major challenge to some of the conclusions the authors draw? Could the authors comment on this? I also did not find a reference to this figure/analysis in the main manuscript. Could the authors add that to the main manuscript.

Response to the Reviewers of Communications Biology COMMSBIO-24-1476, "Transient destabilization of whole brain dynamics induced by DMT"

We thank the Reviewers for carefully reading our work and for sharing their insightful constructive criticism, which led to substantial changes to the text and figures of our manuscript, including the addition of new supplementary material. We hope that our submission has improved as a result of the efforts we invested to address these concerns.

Please see below our point-by-point response to these concerns (Reviewer comments are in red).

Reviewer #1

1. It would be great to see if reactivity would differ for within RSN connectivity versus inter-RSN connections (i.e., to separate the intra and inter-network connections in the FCD matrices).

We thank the Reviewer for this interesting suggestion. To compute the inter-network FCD, we grouped the time series of the nodes corresponding to each network, and afterwards we averaged them to obtain a mean time series per RSN. This resulted in six time series that were used to compute an FCD matrix. Given that the perturbations were introduced at each of these RSN independently, this resulted in six perturbed FCD matrices which were used to calculate the reactivity to the perturbation, as explained in the Methods section. Figure R1 (see Supplementary Figure 9) shows a plot analogous to Figure 4 of the main manuscript text, but only for the inter-network contribution to the FCD. It can be observed that even though reactivity for DMT is higher than for the placebo, the overall temporal structure is lost. Combined with the results shown in the main manuscript text, Figure R1 suggests that the effects of DMT are less manifest in the dynamics of inter-RSN functional connectivity. Supplementary Figure 9 is referenced in the Results section of the revised manuscript when presenting the results corresponding to Figure 4.

Concerning the suggestion to repeat this analysis for inter-RSN connections, while we believe that this is a valuable suggestion, we are concerned with the number of analyses to be performed and how their presentation and subsequent discussion may represent a significant detour from the main findings of our work. This is because there are 15 RSN pairs, so that the between RSN analysis consists of replicating the findings of Figure R1 (equivalently, Figure R4 of the main manuscript) 15 times. We believe the exhaustive exploration of these analysis would require a sufficient amount of space and effort to merit an independent inquiry.

Figure R1. Time-dependent effects of simulated perturbations indicate higher reactivity for DMT vs. placebo for inter-network connections. To compute the inter-network FCD, the time series of the nodes corresponding to each network were grouped, and afterwards were averaged to obtain a mean time series per RSN. This resulted in six time series that were used to compute an FCD matrix. Given that the RSNs were perturbed independently, this resulted in six perturbed FCD matrices which were used to calculate the reactivity. In the left panel we show the reactivity normalized by the number of nodes in the corresponding resting state network (RSN), for placebo (top) and DMT (bottom). Shaded regions of each line denote the standard deviation of the bootstrapped reactivities ($n_{boot} = 200$). Left panel depicts the plots of the peak $\chi(t)$ across time (χ_{max}) for each RSN and three different external perturbation intensities (F_{ext}) (DMT: dimethyltryptamine).

2. Page 21, please correct the error in the model equation 1, that is xn.

We thank the Reviewer for detecting this typo. It will be corrected in the revised version of our manuscript.

3. Global coupling parameter G was set to 0.5. While the authors mention that this was optimized for another cohort in (Ipina et al., 2020), this might not be an optimal choice given several major differences, including for example:

- The cohort and conditions are largely different
- Due to the difference in estimating structural connectivity measures and also max normalization of structural connectome to 0.2 in the previous study.

Still, this would be acceptable if the authors could report the model's performance (for example, (dis)similarity of connectivity measures, at the optimal working point of model (i.e., for the chosen parameter sets).

We thank the Reviewer for this observation. Even though we referenced a previous study, we also performed a pilot set of simulations to confirm the choice of the coupling parameter, finding that $G=0.5$ resulted in optimal goodness of fit values. This information was added to the revised manuscript (please see below). Nevertheless, we agree with the Reviewer's recommendation that it would be informative to report the model performance at the chosen coupling parameter. For this purpose, we added to the supplementary material the grid search corresponding to the aforementioned pilot simulations (Supplementary Figure 1), incorporating additional simulations to obtain more

representative values. This figure is reproduced here as Figure R2A. It is clear from this plot that an extended region of parameter space resulted in optimal goodness of fit values. Even though the average global minimum in the parameter space (Figure R2A, red star) was not the one used in our manuscript (Figure R2A, yellow star), we nevertheless observed that choosing $G=0.5$ resulted in a point within the same extended optimal region. Importantly, no significant difference in the goodness of fit was observed between $G=0.5$ and the global minimum (bootstrapped over 1000 samples). Figure R2B shows that the distance between empirical and simulated data reaches a minimum at $a=0.07$ with a value ≈ 0.086 for $G = 0.5$ (yellow curve), which is almost identical to the minimum obtained at $a \approx 0.1$ for $G=1.9$. Since according to these analyses both choices result in equivalent model performance, we decided to set $G=0.5$ as this choice is consistent with our previous work.

We included a discussion about these points in the revised Methods, subsection “Model fitting to baseline data”:

“To fit the model to the baseline data before drug administration, we first conducted a search for the optimal values of the bifurcation parameter a and the coupling parameter G . This was done by fitting the FCD submatrix comprising the 22 temporal windows (corresponding to the first 8 minutes before DMT injection) averaged over all subjects during both sessions (30 submatrices in total). We performed an exhaustive parameter space exploration by varying the two free parameters of the model: a was changed homogeneously across all nodes from -0.1 to 0.1 in increments of 0.01 , while G was varied from 0 to 2 in steps of 0.1 . This procedure was performed 30 times for each pair of parameters, and the resulting distance metrics were then averaged to determine the optimal parameters (see Supplementary Figure 1A). We observed that the optimal values spanned an extended region in parameter space, which included $G = 0.5$, a value consistent with previous determinations of the coupling parameter in similar datasets (Ipina et al., 2020). While the global minimum was found for another value of G , the resulting goodness of fit was almost identical to the value obtained for $G = 0.5$, without significant differences between both choices (see panel B of Supplementary Figure 1). With the choice of $G=0.5$ and $a=0.07$, the Euclidean distance found for the baseline was 0.084 ± 0.005 , 95% confidence interval [CI]”

Figure R2. A) Exhaustive exploration of the 2D parameter space for the Hopf model applied to the baseline condition (i.e. before DMT infusion), averaged over 30 simulations for each parameter pair. The colorbar indicates the Euclidean distance between the simulated and empirical FCD matrices. The global minimum and the minimum corresponding to $G=0.5$ are marked with a red and yellow star, respectively. **(B)** Euclidean distance as a function of the bifurcation parameter for the two coupling parameters indicated in panel A. The inset shows the distribution of Cohen's d effect size estimate between both choices of model parameters, obtained using a bootstrap procedure with $n = 1000$ repetitions. No significant difference in the goodness of fit was found between the two model parameter values (DMT: dimethyltryptamine).

4. In Figure 2:

- Please consider revising Figure title, given that exploration of parameter space is not about “optimal Functional Connectivity Dynamics”, but optimal parameters for model fitting.
- “Exploration of parameter space and optimal Functional Connectivity Dynamics”
- Please also add a label to the colorbars.

- Please remove the diagonal lines in Figure 2B, and accordingly update the color bar. Having a more focused range of values in this plot and its respective colorbar, would be beneficial for seeing any potential pattern. It would be good to also add region labels.
- Also, please mark chosen parameter values in the plot, and report the exact (dis)similarity of connectivity measures (i.e., the fitting cost) for each chosen parameter set.

We thank the Reviewer for these suggestions. First, we would like to note that the title of the figure stems from what is represented in both of its panels. Panel A contains the exhaustive exploration of parameter space while panel B compares the simulated vs. empirical Functional Connectivity Dynamics (FCD) for the optimal model parameters, hence the title “*Exploration of parameter space and optimal Functional Connectivity Dynamics*”. However, we understand that this could be confusing and therefore we opted to clarify this in the revised version of the manuscript, changing the figure title to “*Exploration of parameter space and Functional Connectivity Dynamics (FCD) corresponding to the optimal model parameters*”.

We added labels to the color bars, as requested by the Reviewer. We also removed the diagonal elements of the matrices shown in panel B. However, since the relatively large matrix entries did not only appear at the main diagonal but also above and below it (indicating high correlation between temporally adjacent FC matrices), we also removed these two other diagonals as well. Finally, we marked the choice of optimal model parameters in panel A. Concerning the request for region labels, we note that the rows and columns of the FCD matrix correspond to different times, not anatomical regions, as is the case of the FC matrix. The new Figure 2 obtained as a result of these changes is reproduced here as Figure R3.

Figure R3. Exploration of parameter space and Functional Connectivity Dynamics (FCD) corresponding to the optimal model parameters. **A)** Normalized Euclidean distance between simulated and empirical FCD averaged across $n = 50$ simulations for every pair of parameters λ and β . The matrices reveal different peak amplitude (λ) and latency (β) values for placebo vs. N,N-Dimethyltryptamine (DMT). Optimal performance for DMT is restricted to a narrower region. The red stars indicate the optimal pair of parameters selected for each condition. **B)** Empirical and optimal simulated FCD (columns) for the placebo and DMT conditions (rows) averaged over $n = 15$ subjects (independent simulations). Simulated FCD matrices were computed using optimal λ and β parameters marked with the red stars in the left panel. Euclidean distances between simulated and empirical FCDs were 0.19 ± 0.03 and 0.14 ± 0.02 (95% confidence interval [CI]) for DMT and placebo respectively (FCD: functional connectivity dynamics; DMT: dimethyltryptamine; DTI: diffusion tensor imaging).

5. Considering that the authors have mentioned the mapping of AAL to the ICA-based RSNs from the Beckman group, it would be beneficial to include a table showing the mapping between AAL regions and RSNs. For example, this table could clarify which regions were labeled as part of the

frontoparietal network. Additionally, more detailed information about this mapping would be appreciated. Given that AAL has relatively low resolution and is anatomical in nature, the choice of RSN labels for some regions might be challenging, particularly when distinguishing between executive and default mode networks.

Thanks for this suggestion. We believe that due to the size of this table, it would not be practical to include it within a text document. Therefore, to facilitate its visualization we have added the requested information to the Github associated with this publication (<https://github.com/juanpiccinini/DMT-whole-brain.git>) where it can be found in .png format.

6. Model fitting was based on a sliding window approach for dynamic functional connectivity (FC). Please provide a (dis)similarity matrix related to the optimization of the bifurcation parameter ‘a’. Additionally, it would be beneficial if the authors could provide both static and dynamic functional connectivity distances between the empirical and simulated data at the model's optimal working point (i.e., for the set of parameters selected after the grid search).

We have added the requested information to the supplementary material of the revised version of our manuscript. The plot of the optimization of parameter a is shown in panel B of Supplementary Figure 1 (reproduced here as Figure R2B), as commented in a previous answer to the Reviewer. Supplementary Figure 2 (shown here as Figure R4) displays the Euclidean distance between the FCD and FC of the empirical data and those obtained from the model. We have referenced this new supplementary figure in the revised Methods section (“*Model fitting to empirical data*” subsection):

“Supplementary Figure 2 presents a complementary comparison of the normalized Euclidean distance and the Pearson correlation computed between the empirical and simulated FCD and FC matrices.”

It is important to indicate that previous research^{1 2} shows that nonlinear oscillator models (such as the Hopf model) may fail to simultaneously optimize multiple targets/metrics - for instance, if a model is tuned to optimize the FCD, it may result in a relatively poor optimization of the static FC, and vice-versa. The results shown in Figure R4 are consistent with this observation. While we observe a relatively high value of the Pearson correlation between empirical and simulated FCD (even though the model was tuned to reproduce the Euclidean distance), the metrics applied to the static FC matrices are comparatively lower. Since our study aimed to capture the dynamic nature of the transition induced by DMT, we opted to use the FCD as an optimization target.

¹ Piccinini, J., Ipiřna, I. P., Laufs, H., Kringelbach, M., Deco, G., Sanz Perl, Y., & Tagliazucchi, E. (2021). Noise-driven multistability vs deterministic chaos in phenomenological semi-empirical models of whole-brain activity. *Chaos: An Interdisciplinary Journal of Nonlinear Science*, 31(2).

² Deco, G., Kringelbach, M. L., Jirsa, V. K., & Ritter, P. (2017). The dynamics of resting fluctuations in the brain: metastability and its dynamical cortical core. *Scientific reports*, 7(1), 3095.

Figure R4. Violin plots of the normalized Euclidean distance (upper panel) and Pearson correlation (bottom panel) between the empirical and simulated Functional Connectivity Dynamics (FCD) and static Functional Connectivity (FC). As explained in the main manuscript text, model optimization was performed using the Euclidean distance as the metric and the empirical FCD as the target. (PCB: Placebo; DMT: dimethyltryptamine).

7. Considering the vulnerability of window-based amplitude correlation (e.g., Pearson measure) to noise, movement artifacts, and choice of window size, instantaneous phase-based measures offer a more reliable assessment of dynamic connectivity. It would be beneficial to include an instantaneous phase-based measure, such as global metastability or synchrony (Kaboodvand et al., 2023: <https://doi.org/10.1371/journal.pcbi.1010958>). The aforementioned study found that the subgroup of patients with higher responsiveness to external neurostimulation were indeed showing blunted frequency dynamics at the baseline, as indexed by lower global metastability and synchrony. Relatedly, it would be interesting to see how global metastability/synchrony would change across time in a plot similar to Figure 4, but extended to the baseline.

We thank the Reviewer for this valuable suggestion. As mentioned by the Reviewer, instantaneous phase-based measures, such as coherence, are an alternative to window-based measures to capture the dynamics of brain functional connectivity. However, some key aspects of our study led us to adopt window-based measures, as explained below.

First, we note that synchrony and metastability are defined as the time average and standard deviation of the Kuramoto order parameter, which reflects the instantaneous level of synchronization between a set of oscillators. However, our objective was not to characterize the time-averaged behavior of functional connectivity (as in the examples provided by the Reviewer). Instead, we sought to investigate how the temporal evolution of whole-brain functional connectivity aligned with DMT pharmacodynamics, and to determine the relationship between this alignment and the dynamics of network stability. Nevertheless, in Supplementary Figure 10 (shown here as Figure R5) we present an

exhaustive exploration of these metrics as a function of the relevant model parameters. Following previous work³, these metrics were defined in terms of relative change, i.e.:

$$M = \frac{o_{sim} - o_{emp}}{o_{emp}}$$

where M represents the distance metric, o_{sim} indicates either the synchrony or metastability obtained from the simulated data, and o_{emp} denotes the corresponding empirical values. As can be seen in this figure, these metrics were unable to capture major differences between conditions. Supplementary Figure 10 is referenced in the Methods section (“*Model fitting to empirical data*” subsection).

Figure R5. Synchronization and metastability between simulated and empirical data averaged across $n = 50$ independent simulations for every pair of model parameters λ and β . Both metrics were calculated as a relative distance between the corresponding simulated and the empirical observable. (PCB: Placebo; DMT: dimethyltryptamine).

³ Piccinini, J., Ipiñna, I. P., Laufs, H., Kringelbach, M., Deco, G., Sanz Perl, Y., & Tagliazucchi, E. (2021). Noise-driven multistability vs deterministic chaos in phenomenological semi-empirical models of whole-brain activity. *Chaos: An Interdisciplinary Journal of Nonlinear Science*, 31(2).

Concerning the use of phase coherence to compute the FCD, we believe that some characteristics of the fMRI time series favor the application of windowed Pearson correlation. Phase coherence is well-defined for narrow-band signals, i.e. signals with non-zero spectral power within a limited part of the spectrum. In other words, coherence is generally applied to estimate the coupling between *oscillations*. In contrast, previous studies suggest that the resting state fMRI signals carry significant information within a more extended range of frequencies, therefore being characterized as *fluctuations*^{4, 5}. In agreement with these observations, we decided to base our analyses on correlations computed over short time windows, as in the original formulation of FCD by Deco and colleagues⁶.

Regardless of this observation, we investigated the agreement between both metrics to provide further support for our methodology. Figure R6 (see also Supplementary Figure 3) presents the correlation between FC matrices computed using phase coherence and sliding windowed Pearson correlation. While some fluctuations are observed across the duration of the scan, we conclude that this correlation is always above 0.6, and above 0.7 for the majority of the time points. This result supports the consistency between both approaches to compute the dynamic coupling between time series.

Figure R6. Correlation between window-based FC and the coherence matrices. The bold line denotes the mean value of the correlation between the FC matrix and the coherence, computed for every time point within the same temporal window, i.e. each value in the plot is the average between each window-based FC and all the coherence matrices within that time window. Shaded regions indicate the standard deviation from the mean. (PCB: Placebo; DMT: dimethyltryptamine; FC: Functional Connectivity).

⁴ Niazy, R. K., Xie, J., Miller, K., Beckmann, C. F., & Smith, S. M. (2011). Spectral characteristics of resting state networks. *Progress in brain research*, 193, 259-276.

⁵ Yuen, N. H., Osachoff, N., & Chen, J. J. (2019). Intrinsic frequencies of the resting-state fMRI signal: the frequency dependence of functional connectivity and the effect of mode mixing. *Frontiers in Neuroscience*, 13, 900.

⁶ Deco, G., Kringelbach, M. L., Jirsa, V. K., & Ritter, P. (2017). The dynamics of resting fluctuations in the brain: metastability and its dynamical cortical core. *Scientific reports*, 7(1), 3095.

Finally, if we use the coherence matrices to compute the FCD, we obtain results that are very similar to those obtained using the sliding-window Pearson correlation (see Supplementary Figure 4, reproduced here as Figure R7), as expected from the results provided in Figure R6. This figure is referenced in the revised Methods section (“Model fitting to empirical data” subsection):

“In Supplementary Figure 4, we compare the obtained FCD matrices with an alternative construction based on phase coherence.”

Figure R7. Empirical FCD matrices computed using coherence between fMRI time series. The i,j element of each matrix denotes the correlation between the coherence matrices associated to times i and j respectively.

8. Empirical data was band-passed 0.01-0.08, where the common upper band in resting-state fMRI analysis is 0.1 or even 0.12. Is that because of limitations of the normal hopf model for producing temporal dynamics? Please provide an illustration of distribution of nodal peak frequencies, that is ω_n , for empirical and simulated data. This is to ensure the normal hopf model has been able to produce comparable temporal dynamics.

We thank the Reviewer for this comment. The BOLD time series were filtered with a second-order Butterworth filter in the range between 0.01 and 0.08 Hz, in line with previous studies.^{7 8 9}. The

⁷ Gu, Y., Lin, Y., Huang, L., Ma, J., Zhang, J., Xiao, Y., ... & Alzheimer's Disease Neuroimaging Initiative. (2020). Abnormal dynamic functional connectivity in Alzheimer's disease. *CNS neuroscience & therapeutics*, 26(9), 962-971.

⁸ Deco, G., Perl, Y. S., Ponce-Alvarez, A., Tagliazucchi, E., Whybrow, P. C., Fuster, J., & Kringelbach, M. L. (2023). One ring to rule them all: the unifying role of prefrontal cortex in steering task-related brain dynamics. *Progress in neurobiology*, 227, 102468.

⁹ Cabral, J., Vidaurre, D., Marques, P., Magalhães, R., Silva Moreira, P., Miguel Soares, J., ... & Kringelbach, M. L. (2017). Cognitive performance in healthy older adults relates to spontaneous switching between states of functional connectivity during rest. *Scientific reports*, 7(1), 5135.

choice of the low-pass cutoff was justified as it helps to filter out physiological noise, which tends to dominate higher frequencies.¹⁰. We have explicitly stated this in the revised version of our manuscript:

“The BOLD time series were filtered with a second-order Butterworth filter in the range between 0.01 and 0.08 Hz, in line with previous studies^{11 12 13 14}. The choice of the low-pass cutoff was justified as it helps to filter out physiological noise, which tends to dominate higher frequencies¹⁵.”

In Supplementary Figure 5 (reproduced here as Figure R8), we display the histogram of the empirical frequencies of the regional fMRI time series. The dotted vertical line indicates the convergence frequency of the simulated local dynamics at the optimal choice of model parameters. This figure shows that local dynamics engage in limit cycle behavior due as a consequence of the positive value of the bifurcation parameter, and that the frequency of the limit cycles is in agreement with the average value of the empirical frequencies. This figure is referenced in the revised Methods section, “Whole-brain computational model”, after introducing the model equations and discussing the relevant model parameters.

¹⁰ Zou, Q. H., Zhu, C. Z., Yang, Y., Zuo, X. N., Long, X. Y., Cao, Q. J., ... & Zang, Y. F. (2008). An improved approach to detection of amplitude of low-frequency fluctuation (ALFF) for resting-state fMRI: fractional ALFF. *Journal of neuroscience methods*, 172(1), 137-141.

¹¹ Zou, Q. H., Zhu, C. Z., Yang, Y., Zuo, X. N., Long, X. Y., Cao, Q. J., ... & Zang, Y. F. (2008). An improved approach to detection of amplitude of low-frequency fluctuation (ALFF) for resting-state fMRI: fractional ALFF. *Journal of neuroscience methods*, 172(1), 137-141.

¹² Cabral, J., Vidaurre, D., Marques, P., Magalhães, R., Silva Moreira, P., Miguel Soares, J., ... & Kringelbach, M. L. (2017). Cognitive performance in healthy older adults relates to spontaneous switching between states of functional connectivity during rest. *Scientific reports*, 7(1), 5135.

¹³ Deco, G., Perl, Y. S., Ponce-Alvarez, A., Tagliazucchi, E., Whybrow, P. C., Fuster, J., & Kringelbach, M. L. (2023). One ring to rule them all: the unifying role of prefrontal cortex in steering task-related brain dynamics. *Progress in neurobiology*, 227, 102468.

¹⁴ Gu, Y., Lin, Y., Huang, L., Ma, J., Zhang, J., Xiao, Y., ... & Alzheimer's Disease Neuroimaging Initiative. (2020). Abnormal dynamic functional connectivity in Alzheimer's disease. *CNS neuroscience & therapeutics*, 26(9), 962-971.

¹⁵ Zou, Q. H., Zhu, C. Z., Yang, Y., Zuo, X. N., Long, X. Y., Cao, Q. J., ... & Zang, Y. F. (2008). An improved approach to detection of amplitude of low-frequency fluctuation (ALFF) for resting-state fMRI: fractional ALFF. *Journal of neuroscience methods*, 172(1), 137-141.

Figure R8. Histogram of empirical frequencies of regional fMRI time series that were used as input for the simulations. The dotted vertical line indicates the convergence frequency of the simulated local dynamics at the optimal choice of model parameters, which agrees with the average value of the empirical frequencies (fMRI: functional magnetic imaging).

9. Please mention if any filtering was applied to the simulated set of signals.

In the case of the simulated data, the temporal series were not filtered given that the frequency of each node was already defined based on the empirical data. Furthermore, given that the optimal bifurcation parameters were positive, the simulated time series engaged in oscillating behavior with frequencies within the range of the empirical values (see Figure R8). This was clarified in the revised version of the Methods section, “Whole-brain computational model” subsection:

“The simulated time series were not band-pass filtered, given that the frequency of each node was defined based on the empirical data, and that the positive optimal bifurcation parameters resulted in oscillating behavior with frequencies within the range of the empirical values.”

10. It’d be great if the authors could share the main model-related scripts, to clarify the details such as the choice of temporal integration method for the differential equation, characterization of noise in the model as well as random initialization method for reproducibility of simulations.

We thank the Reviewer for the suggestion. We have added the requested information to the repository of the manuscript <https://github.com/juanpiccinini/DMT-whole-brain>. This information has been added to the revised version of the manuscript (“Whole-brain computational model” subsection).

11. In page 5, lines 118 to 121, a big body of previous whole-brain modeling studies which have used dynamic connectivity measures have been glossed over. For example, (Deco et al., 2017; Kaboodvand et al., 2019)

Thanks for this suggestion. We cited these works in the revised version of our manuscript.

12. Please add the list of all abbreviations in figure captions.

We have added the requested information to the revised version of our manuscript.

13. Please mention the power limitation of this study, including the exact number of participants used in this study (that is n=15), and the number of subjects in active versus placebo groups.

Thanks for this suggestion. We have added the requested information to the revised version of our manuscript. In particular, the potential power limitation is mentioned as a limitation in the revised Discussion section. Given that all subjects participated in both conditions, the number of participants in the active and placebo groups is the same. The discussion added to the revised manuscript is as follows:

“Another limitation is the number of participants (n=15), which is comparatively low for a pharmacological fMRI study. In the case of this study, after applying strict criteria to exclude subjects due to head motion in the scanner only 75% of the original data was included. While we modeled group-averaged FCD matrices, future studies at the single subject level should attempt to raise the effective number of subjects included in the model.”

14. Regarding the demographic details, it would be more informative to be reported for n=15 included subjects, rather than the initial n=20 sample. For example, page 18 says that: “A cohort of 20 participants completed all study visits, consisting of 7 females with a mean age of 33.5 years and a standard deviation of 7.9.”

Thanks for this suggestion. We have added the requested information to the revised version of our manuscript (“Study participants and experimental design” subsection).

15. Could you please report the similarity between structural and static functional connectivity of baseline, for both empirical and simulated BOLD?

We have added the requested information to the supplementary material of the revised version of our manuscript, reproduced here as Figure R9. This figure was incorporated as Supplementary Figure 6 and referenced in the revised Methods section, at the end of the “Anatomical connectivity matrix” subsection:

“Supplementary Figure 6 summarizes the similarity between structural and functional connectivity, both for simulated and empirical data.”

Figure R9. Pearson correlation between the simulated Functional Connectivity (FC) and the Structural Connectivity (SC) for the baseline period, i.e. the FC matrices were calculated for the time before the infusion of DMT or placebo. The histograms were obtained across independent simulations of the model and for every subject of the study in each condition. The average correlation between FC and SC is 0.303 ± 0.008 and 0.30 ± 0.06 (95% confidence interval), for the simulations and subjects respectively (DMT: dimethyltryptamine).

16. Was the diffusion tensor imaging data collected from the same sample? Otherwise, it would be more reliable to use the structural connectome from larger samples, such the human connectome project. Please add more details about the construction of group-average structural connectome and applied normalizations. In case the script for estimation of connectivity probabilities and following postprocessing have been included in the shared directory, it's good to refer to the relevant script in the anatomical connectivity matrix section, page 20.

Since diffusion tensor imaging (DTI) data was not available for the group of subjects who participated in this study, we computed the structural connectivity (SC) matrix using tractography data obtained from an independent sample. Instead of computing the SC matrix from publicly available data, we decided to use a sample of high quality data scanned by co-authors of this manuscript using an optimized high-resolution sequence. Moreover, this data was processed using the same pipeline chosen for previous whole-brain modeling studies, to increase the consistency and interpretability of our results within the context of our previous work. The information concerning the acquisition and processing of the DSI data to construct the SC matrices is provided below, and included in the revised version of the Supplementary Material under the section "Structural Connectivity: DWI data collection and processing".

17. Given that the normal Hopf model is particularly well-suited for simulating the effects of external manipulation (including perturbation and neurostimulation, e.g. (Spiegler et al., 2016, Kaboodvand et al., 2019, Iravani et al., 2021), it would be good to discuss the perturbation choice of this study and similarities or advantages to alternative ways.

Thank you for this suggestion. In our study, we stimulated the dynamics using a periodic signal matching the frequency of the local dynamics, i.e. the resonant frequency. The resonant frequency is a natural choice given that it is known to elicit a maximal response (see for instance Figure 5 of Jobst et al., 2017¹⁶). Moreover, this is a natural choice since more complex signals can be represented in the frequency space via a Fourier decomposition, and in that case the response will be predominantly elicited by the amplitude of the Fourier component at the resonant frequency.

We note that the papers suggested by the Reviewer adopted a different form of stimulation, consisting of an impulse in the time domain which models a Transcranial Magnetic Stimulation (TMS) pulse. In our model, this corresponds to displacing the dynamics away from the limit cycle and determining the necessary time to recover the baseline dynamics. This choice presents the advantage of representing the effects of a TMS pulse localized in time, one of the most commonly used forms of transcranial stimulation. However, this form of stimulation is not localized in the frequency domain, and therefore it does not inform how the model responds to specific frequencies. We opted to pursue the latter alternative, however we briefly discuss these other possibilities in the revised version of our work in the revised version of the Discussion:

“We opted to simulate perturbations of the ongoing dynamics with a periodic signal matching the frequency of the local dynamics, i.e. the resonant frequency. This choice represents transcranial alternating current stimulation (tACS) at the peak frequency of the endogenous oscillations. The resonant frequency is a natural choice for stimulation, given that it is known to elicit a maximal response¹⁷. Also, more complicated signals can be represented in the frequency space via their Fourier decomposition, where the response will be predominantly elicited by the amplitude of the Fourier component at the resonant frequency. Previous studies modeled other forms of stimulation resembling a single transcranial magnetic stimulation (TMS) pulse^{18 19 20}. In our model, this would correspond to displacing the dynamics away from the limit cycle and then determining the necessary time to recover baseline dynamics. This choice presents the advantage of representing the effects of a TMS pulse localized in time, one of the most commonly used forms of transcranial stimulation. However, it is unlocalized in the frequency domain, and therefore it does not inform how the model responds to specific frequencies.”

18. Minor:

¹⁶ Jobst, B. M., Hindriks, R., Laufs, H., Tagliazucchi, E., Hahn, G., Ponce-Alvarez, A., ... & Deco, G. (2017). Increased stability and breakdown of brain effective connectivity during slow-wave sleep: mechanistic insights from whole-brain computational modelling. *Scientific reports*, 7(1), 4634.

¹⁷ Jobst, B. M., Hindriks, R., Laufs, H., Tagliazucchi, E., Hahn, G., Ponce-Alvarez, A., ... & Deco, G. (2017). Increased stability and breakdown of brain effective connectivity during slow-wave sleep: mechanistic insights from whole-brain computational modelling. *Scientific reports*, 7(1), 4634.

¹⁸ Spiegler, Andreas, et al. "Selective activation of resting-state networks following focal stimulation in a connectome-based network model of the human brain." *eneuro* 3.5 (2016).

¹⁹ Kaboodvand, Neda, Martijn P. van den Heuvel, and Peter Fransson. "Adaptive frequency-based modeling of whole-brain oscillations: Predicting regional vulnerability and hazardousness rates." *Network Neuroscience* 3.4 (2019): 1094-1120.

²⁰ Iravani, Behzad, et al. "Whole-brain modelling of resting state fMRI differentiates ADHD subtypes and facilitates stratified neuro-stimulation therapy." *Neuroimage* 231 (2021): 117844.

'Fig' and 'Figure' have been both used interchangeably. Please keep consistency.

Page 20, Stuard-Landau -> Stuart-Landau

Since we did not find explicit instructions in the journal website, our article was written to conform to the APA style guidelines, which discourage the use of acronyms or abbreviations at the beginning of a new sentence (see for instance <https://writing.wisc.edu/handbook/docapa/docapaprinciples/>). Thus, we used "Figure" at the beginning of a new sentence and/or caption, and "Fig." elsewhere. While eventually this can be adjusted by the journal based on its own style guidelines, we followed the suggestion of changing all instances of "Fig." to "Figure".

Reviewer #2

1. After the model was fitted to empirical data (FCD) during parameter optimization, it is supposed to reflect the brain dynamics during the drug administration. I however missed convincing estimations of how close the modeling results replicate the empirical observations.

I appreciate that the FCDs were compared based on the Frobenius distance, which is supposed to better reflect the closeness of FC dynamics, and which would be missed otherwise for a statistical comparison of the FCD distributions. But it is difficult to interpret the obtained distances (Fig.2). Calculating the correlation between the empirical and simulated FCDs may better reflect the pattern similarity and provide an evidence of how similar they are to each other in DMT and placebo cases.

We thank the Reviewer for raising this important discussion. As stated by the Reviewer, we followed previous works^{21 22 23} in the use of Euclidean distance (also known as Frobenius distance) to optimize the model parameters. Other metrics are available, each presenting distinct advantages and potential drawbacks. Even though the Pearson correlation may better reflect the similarity between the structure of the empirical and simulated FCD matrices, it may fail to account for the magnitude of such fluctuations; i.e., the mean value of the simulated matrix could be off by far with respect to the empirical ones and the correlation will not capture that. To provide additional robustness against the potential limitations of the Frobenius distance, we normalized it by dividing by the norm of the empirical FCD, so that the results can be interpreted as the relative variation of the simulation with respect to the empirical data. This point is now explicitly stated in the revised version of our manuscript in the "*Fitting the temporal evolution of the bifurcation parameter*" subsection:

"These matrices were computed for each of the fifteen participants and simulations by exhaustively exploring the model parameters related to the temporal evolution of the bifurcation parameter, λ and β . To compare the FCD matrices taking into account their temporal structure, we used the Euclidean distance between the elements of the empirical and simulated matrices, and then we normalized the results dividing by the empirical norm to account for the relative change."

²¹ Jobst, B. M., Hindriks, R., Laufs, H., Tagliazucchi, E., Hahn, G., Ponce-Alvarez, A., ... & Deco, G. (2017). Increased stability and breakdown of brain effective connectivity during slow-wave sleep: mechanistic insights from whole-brain computational modelling. *Scientific reports*, 7(1), 4634.

²² Piccinini, J., Ipiñna, I. P., Laufs, H., Kringelbach, M., Deco, G., Sanz Perl, Y., & Tagliazucchi, E. (2021). Noise-driven multistability vs deterministic chaos in phenomenological semi-empirical models of whole-brain activity. *Chaos: An Interdisciplinary Journal of Nonlinear Science*, 31(2).

²³ Sastry, N. C., Roy, D., & Banerjee, A. (2023). Stability of sensorimotor network sculpts the dynamic repertoire of resting state over lifespan. *Cerebral Cortex*, 33(4), 1246-1262.

However, following the suggestion made by the Reviewer, in the new Supplementary Figure 2 (reproduced here as Figure R4) we have computed the correlation between the empirical and simulated FCD matrices as a complementary metric to inform the similarity between them. Following the suggestions made by Reviewer #1, we also added the similarity of the FC matrices to this figure. In this figure, we observe a relatively high value of the Pearson correlation between empirical and simulated FCD, even though the model was tuned to reproduce the Euclidean distance.

Because of the well-defined reference time point of drug administration, why were the empirical and simulated FC matrices calculated at every sliding window not compared to each other at these time points? Such a comparison could also show how well the model with varied parameter followed the recorded brain activity.

We thank the Reviewer for this observation. Our approach was to summarize the post-DMT/placebo whole-brain FC trajectory using the FCD, with the objective of optimizing the parametrization of $a(t)$ to maximize the similarity between empirical and simulated data. While interesting, the approach suggested by the Reviewer is considerably different, as it results in an individual FC matrix per sliding window. It is unclear how the model parameters could be optimized to simultaneously fit the multiple FC matrices obtained as a result of this procedure. In our work, this complex multi-target optimization problem is avoided by focusing on the FCD matrix, which represents the evolution of similarity between transient brain states, instead of focusing on the brain states themselves.

Another potential issue is the noise introduced by computing FC over short time series. If the model is fitted to the FC matrices computed at each sliding window, this noise could exert a significant effect on the model parameters. Conversely, and as shown in our reply to a concern raised by Reviewer #1 (see Figure R7, added also as the new Supplementary Figure 4), the structure of the FCD matrices is maintained even if the instantaneous phase correlation is employed to determine the matrix elements.

Finally, another reason to prioritize the reproduction of the FCD is the limited capability of nonlinear oscillator models (such as the Hopf model) to simultaneously reproduce multiple targets/metric^{24 25}. For instance, if a model is tuned to optimize the FCD, it may result in a relatively poor optimization of the static FC, and vice-versa. As can be seen in Figure R4, while we observe a relatively high value of the Pearson correlation between empirical and simulated FCD (even though the model was tuned to reproduce the Euclidean distance), the metrics applied to the static FC matrices are comparatively lower. Since our study aimed to capture the dynamic nature of the transition induced by DMT, we opted to prioritize the FCD as the optimization target.

2. Before coming to periodic stimulation of the obtained model with optimized parameters, where the corresponding empirical data are absent as far as I can infer, its output can be compared with other empirical observations available from the studies cited in the manuscript. For example, Ref. [9] demonstrated a few interesting results for the same data used in the current manuscript, where "... fMRI results revealed robust increases in global functional connectivity (GFC), network disintegration and desegregation, and a compression of the principal cortical gradient under DMT."

²⁴ Piccinini, J., Ipiñna, I. P., Laufs, H., Kringelbach, M., Deco, G., Sanz Perl, Y., & Tagliazucchi, E. (2021). Noise-driven multistability vs deterministic chaos in phenomenological semi-empirical models of whole-brain activity. *Chaos: An Interdisciplinary Journal of Nonlinear Science*, 31(2).

²⁵ Deco, G., Kringelbach, M. L., Jirsa, V. K., & Ritter, P. (2017). The dynamics of resting fluctuations in the brain: metastability and its dynamical cortical core. *Scientific reports*, 7(1), 3095.

The comments and suggestions put forward by the reviewer are aligned with the previous point (i.e. point #1), given that the suggested metrics (e.g. global functional connectivity, network disintegration/desegregation, principal cortical gradient compression) depend directly on the structure of the FC matrix. These interesting suggestions are compatible with an attempt to model the time-to-time similarity in the configuration of the FC matrix. Instead, we pursued the alternative objective of modeling the FC dynamics as encoded in the FCD matrix, according to the rationale exposed in our reply to the previous point. As whole-brain models offer a limited capacity to simultaneously capture dynamic (e.g. FCD) and static (e.g. FC) optimization targets^{26 27}, this choice led us to prioritize the fitting of the FCD. The subsequent choice of studying the reactivity upon external perturbations follows directly from this choice, given that FCD was optimized in terms of the parameters that determine the temporal evolution of $a(t)$, i.e. the temporal evolution of the proximity to the bifurcation point, which is directly implicated in the stability of the dynamics when driven by external perturbations.

More generally, we consider that the first step taken in modeling studies is the determination of what specific aspects of the data will be reproduced by the model. This choice can be driven by scientific and/or practical considerations. In our work, we decided to focus on capturing the dynamics of whole-brain FC (as represented by the FCD) in relation to the dynamics of the DMT experience and the drug pharmacodynamics, as opposed to modeling the time-to-time changes in the FC, which could lead to a complex multi-target optimization problem. Given previous reports of links between psychedelics and dynamic criticality^{28 29 30}, the temporal dynamics were introduced in the evolution of the $a(t)$, the bifurcation parameter, which plays a key role in the stability of the dynamics. As our work does not attempt to be exhaustive regarding the choice of models and optimization targets/metrics, there is room for future studies pursuing other avenues of interest, including those mentioned by the reviewer.

By confirming the reported observations for the modeling results could enhance the confidence in the model and its other outcomes, where a direct comparison with empirical data is not possible so far, which may question the respective modeling results with respect to their correspondence to reality. Therefore, the results of study [42] should be discussed in more detail, where in particular was shown that stronger stimuli were accompanied by a weaker differentiation between drug-induced and placebo conditions, which might be contrasted with the presented modeling results.

We thank the suggestion of expanding our discussion of Ref. 42 in our manuscript (Mediano, Pedro AM, et al. "Effects of external stimulation on psychedelic state neurodynamics." *ACS Chemical Neuroscience* 15.3 (2024): 462-471.). Among other interesting results, this work shows that external

²⁶ Piccinini, J., Ipiřna, I. P., Laufs, H., Kringelbach, M., Deco, G., Sanz Perl, Y., & Tagliazucchi, E. (2021). Noise-driven multistability vs deterministic chaos in phenomenological semi-empirical models of whole-brain activity. *Chaos: An Interdisciplinary Journal of Nonlinear Science*, 31(2).

²⁷ Deco, G., Kringelbach, M. L., Jirsa, V. K., & Ritter, P. (2017). The dynamics of resting fluctuations in the brain: metastability and its dynamical cortical core. *Scientific reports*, 7(1), 3095.

²⁸ Girn, M., Rosas, F. E., Daws, R. E., Gallen, C. L., Gazzaley, A., & Carhart-Harris, R. L. (2023). A complex systems perspective on psychedelic brain action. *Trends in Cognitive Sciences*, 27(5), 433-445.

²⁹ Demertzi, A., Tagliazucchi, E., Dehaene, S., Deco, G., Barttfeld, P., Raimondo, F., ... & Sitt, J. D. (2019). Human consciousness is supported by dynamic complex patterns of brain signal coordination. *Science advances*, 5(2), eaat7603.

³⁰ Solovey, G., Alonso, L. M., Yanagawa, T., Fujii, N., Magnasco, M. O., Cecchi, G. A., & Proekt, A. (2015). Loss of consciousness is associated with stabilization of cortical activity. *Journal of Neuroscience*, 35(30), 10866-10877.

stimulation disrupts the difference between LSD and placebo, as determined by the Lempel-Ziv entropy of brain activity time series. To establish this result, different conditions were examined, in order of increasing external stimulation strength: eyes closed, eyes closed with music, eyes open, and eyes open during a video presented to participants. It is important to observe that the reduction in the effect of the stimulation was at least partially driven by increased entropy of the baseline condition, i.e. the placebo condition. Moreover, by contrasting the placebo condition under different forms of stimulation, the authors concluded that the content and structure of the stimulation was linked to the change in brain entropy, since the eyes open condition presented less brain entropy than the eyes open condition with video stimulation.

Our modeling study concluded that external stimulation delivered during the peak of the DMT effects has the strongest effect on the ongoing dynamics relative to the placebo. As expected by the results in Mediano et al., the highly structured and periodic nature of the external stimulation exerted a comparatively weak effect on the baseline (placebo) vs. drug condition. Therefore, the net effect of the stimulation was larger in the drug vs. placebo condition. It is also important to note that using Lempel Ziv entropy to determine the effects of the stimulation is fundamentally different from the approach we used in our manuscript. These considerations have been added to the revised Discussion section of our manuscript:

“It is also important to note that in the study by Mediano et al. the content and structure of the stimulation was linked to the change in brain entropy, and that the results were partially driven by increased entropy of the baseline condition. In contrast, in our analyses the structured and periodic nature of the external stimulation exerted a comparatively weak effect on the placebo vs. drug condition.”

3. Through the entire study the coupling parameter was fixed to $G = 0.5$ and taken from another study [39] dedicated to very different modeling conditions and data. It is not immediately evident why this was justified, especially because this parameter could be important to appropriately fit the model. Furthermore, if the bifurcation parameter $a(t)$ was allowed to vary, why was this denied for the coupling parameter, where they may influence each other and impact the modeling results and conclusions, in particular, "increases in global functional connectivity (GFC)" mentioned above?

We thank the Reviewer for this interesting observation. Even though we referenced a previous study, we also performed a pilot set of simulations to confirm the choice of the coupling parameter, finding that $G=0.5$ resulted in optimal goodness of fit values. This information was added to the revised manuscript (please see below). Nevertheless, we agree with the Reviewer's recommendation to add further support to our choice of this parameter. For this purpose, we added to the supplementary material (Supplementary Figure 1) the grid search corresponding to these pilot simulations, incorporating additional simulations to obtain more representative values, reproduced here as Figure R2A. It is clear from this plot that an extended region of parameter space resulted in optimal goodness of fit values. Even though the average global minimum in the parameter space (Figure R2A, red star) is not the one used in our manuscript (Figure R2A, yellow star), we observed that choosing $G=0.5$ results in a point within the same optimal region. Importantly, no significant difference in the goodness of fit was observed between $G=0.5$ and the global minimum (bootstrapped over 1000 samples). Figure R2B shows that the distance between empirical and simulated data reaches a minimum at $a=0.07$ with a value ≈ 0.086 for $G = 0.5$ (yellow curve), which is almost identical to the minimum obtained at $a \approx 0.1$ for $G=1.9$. Since both choices result in comparable model performance, we decided to set $G=0.5$ as this result is consistent with our previous work.

We included a discussion about these points in the revised Methods, subsection “Model fitting to baseline data”:

“To fit the model to the baseline data before drug administration, we first conducted a search for the optimal values of the bifurcation parameter a and the coupling parameter G . This was done by fitting the FCD submatrix comprising the 22 temporal windows (corresponding to the first 8 minutes before DMT injection) averaged over all subjects during both sessions (30 submatrices in total). We performed an exhaustive parameter space exploration by varying the two free parameters of the model: a was changed homogeneously across all nodes from -0.1 to 0.1 in increments of 0.01, while G was varied from 0 to 2 in steps of 0.1. This procedure was performed 30 times for each pair of parameters, and the resulting distance metrics were then averaged to determine the optimal parameters (see Supplementary Figure 1A). We observed that the optimal values spanned an extended region in parameter space, which included $G = 0.5$, a value consistent with previous determinations of the coupling parameter in similar datasets (Ipiña et al., 2020). While the global minimum was found for another value of G , the resulting goodness of fit was almost identical to the value obtained for $G = 0.5$, without significant differences between both choices (see panel B of Supplementary Figure 1). With the choice of $G=0.5$ and $a=0.07$, the Euclidean distance found for the baseline was 0.084 ± 0.005 , 95% confidence interval [CI]”

Concerning the temporal dependency of the bifurcation parameter (a) instead of the coupling parameter (G), this choice is consistent with several previous work by our team and others^{31 32 33} where the effect of external manipulations is introduced at the local dynamics, and any modifications to the inter-areal coupling emerge as a consequence. Neurobiologically, psychedelic drugs bind locally to cell populations and exert an influence in their biophysical properties, eventually resulting in changes to the level of long-range neural synchronization/desynchronization³⁴. Modeling the action of psychedelics as changes in the bifurcation parameter a also follows from experimental results that suggest these compounds introduce qualitative changes to global brain dynamics, which could be compatible with a transition between super- and sub-critical dynamics³⁵. Finally, modeling changes to the bifurcation parameter emerges as a natural choice when investigating psychedelic-induced changes to dynamic stability, given that in the weak coupling regime, network stability primarily depends on the proximity of local dynamics to the Hopf bifurcation³⁶.

4. Similar question arises from the considering the group-averaged structural connectivity (SC) from different data. If we proceed with such assumptions even further to extreme, then we may claim that

³¹ Ipiña, I. P., Kehoe, P. D., Kringelbach, M., Laufs, H., Ibañez, A., Deco, G., ... & Tagliazucchi, E. (2020). Modeling regional changes in dynamic stability during sleep and wakefulness. *NeuroImage*, 215, 116833.

³² Sanz Perl, Y., Pallavicini, C., Pérez Ipiña, I., Demertzi, A., Bonhomme, V., Martial, C., ... & Tagliazucchi, E. (2021). Perturbations in dynamical models of whole-brain activity dissociate between the level and stability of consciousness. *PLoS computational biology*, 17(7), e1009139.

³³ Deco, G., Cruzat, J., Cabral, J., Knudsen, G. M., Carhart-Harris, R. L., Whybrow, P. C., ... & Kringelbach, M. L. (2018). Whole-brain multimodal neuroimaging model using serotonin receptor maps explains non-linear functional effects of LSD. *Current biology*, 28(19), 3065-3074.

³⁴ Carhart-Harris, R. L. (2019). How do psychedelics work?. *Current opinion in psychiatry*, 32(1), 16-21.

³⁵ Girn, M., Rosas, F. E., Daws, R. E., Gallen, C. L., Gazzaley, A., & Carhart-Harris, R. L. (2023). A complex systems perspective on psychedelic brain action. *Trends in Cognitive Sciences*, 27(5), 433-445.

³⁶ Jobst, B. M., Hindriks, R., Laufs, H., Tagliazucchi, E., Hahn, G., Ponce-Alvarez, A., ... & Deco, G. (2017). Increased stability and breakdown of brain effective connectivity during slow-wave sleep: mechanistic insights from whole-brain computational modelling. *Scientific reports*, 7(1), 4634.

there exists one SC (and one value of the coupling parameter) that can be considered for modeling whatever (healthy) functional data we take for whatever modeling conditions, which is confusing and difficult to accept.

In the case of our study, the use of group-averaged SC is motivated by lack of subject-specific tractography data, and justified by the observation that psychedelics do not modify anatomical connections within the duration of the acute effects and after a single administration of the drug. While psychedelics may introduce structural changes to the brain via their effects on neural plasticity, current evidence points towards longer time scales and administration schedules that differ from the experimental setting of our work³⁷.

Indeed, taking this justification to the extreme and concluding that the same SC and coupling parameter can be used for modeling whatever functional data could be difficult to accept. This may be so, as indeed there are some conditions characterized by structural dysconnectivity deficits which likely require subject or group-specific SC to be modeled, and/or changes to the coupling parameter G ³⁸. However, the point is not whether our decision can be justified for modeling any population of subjects; the point is whether this justification exists and is valid for the case of brain effects induced by DMT, which is the objective of our study.

One limitation of employing a group-average estimate of the SC is that the resulting models fail to include between-subject variability. To overcome the lack of DTI data for the group of subjects who participated in this study, we decided to use a sample of high quality data scanned by co-authors of this manuscript using an optimized high-resolution Diffusion Tensor Imaging (DTI) sequence, processed using the same pipeline chosen for previous whole-brain modeling studies to increase the consistency and interpretability of our results within the context of our previous work (see the new subsection “Structural Connectivity: DWI data collection and processing” of the Supplementary Material for detailed information concerning the acquisition and processing of the DTI data). This approach is frequently adopted when single-subject DTI data is not available, and the condition being modeled is not related to disruptions in long-range white matter fiber tracts.^{39 40 41}

5. The manuscript also investigated the sensitivity of the model to perturbations, where a model reactivity was calculated. Unfortunately, the methodological approach (lines 516-527) is very unclear and confusing, and should be better explained and justified.

It is not clear what the "equally spaced values of $a(t)$ " are. Then, the simulations appeared to be performed for constant values of the parameter, $a=0.07$ for the pre-dose time interval and for some

³⁷ De Vos, C. M., Mason, N. L., & Kuypers, K. P. (2021). Psychedelics and neuroplasticity: a systematic review unraveling the biological underpinnings of psychedelics. *Frontiers in psychiatry*, 12, 724606.

³⁸ van den Heuvel, M. P., & Sporns, O. (2019). A cross-disorder connectome landscape of brain dysconnectivity. *Nature reviews neuroscience*, 20(7), 435-446.

³⁹ Sanz Perl, Y., Pallavicini, C., Pérez Ipiña, I., Demertzi, A., Bonhomme, V., Martial, C., ... & Tagliazucchi, E. (2021). Perturbations in dynamical models of whole-brain activity dissociate between the level and stability of consciousness. *PLoS computational biology*, 17(7), e1009139.

⁴⁰ Herzog, R., Mediano, P. A., Rosas, F. E., Lodder, P., Carhart-Harris, R., Perl, Y. S., ... & Cofre, R. (2023). A whole-brain model of the neural entropy increase elicited by psychedelic drugs. *Scientific reports*, 13(1), 6244.

⁴¹ Singleton, S. P., Luppi, A. I., Carhart-Harris, R. L., Cruzat, J., Roseman, L., Nutt, D. J., ... & Kuceyeski, A. (2022). Receptor-informed network control theory links LSD and psilocybin to a flattening of the brain's control energy landscape. *Nature communications*, 13(1), 5812.

constant a_p values after the drug administration. We however learned that $a(t)$ should vary with time during this epoch. The transient model dynamics and its (transient) reaction on the perturbations might be different for these two conditions of constant and time-varied a .

"The perturbation was applied after the first 8 minutes corresponding to the baseline, varying the amplitude F_{ext} from 0 to 0.015 in 0.0125 steps." Can I understand this so that the amplitude was varied (increased) with time instead of parameter a ? If true, why not otherwise?

Next, "The maximum value of the range was chosen" - which value, from which range and chosen for what and why? This was done because "it saturated the correlation between 5HT2a receptor density and local reactivity, as shown in Fig. 5". Here I may see some indications of a circular logic, where the information about correlation was used to obtain results that will be employed to demonstrate the correlations.

In Fig.5 we see a non-monotonic behavior of the correlation versus the stimulation intensity with smaller correlation for large/small forcing. The large values of intensity were chosen, if I read it correctly, but again, for what, especially because all combinations of the parameters were then considered: "For each combination of RSN, F_{ext} and a_p we computed the resulting FCD and its distance to the empirical condition"?

We thank the Reviewer for pointing out some deficiencies in our explanation, and for giving us the opportunity to correct them. First of all, regarding the equally spaced values of $a(t)$, indeed, we performed the simulations for constant values of the parameter: $a=0.07$ for the pre-dose time interval and for some constant a_p value after the drug administration. The value of a_p was selected from the optimal gamma function corresponding to each condition, according to the previous fit of the FCD matrix. To me more specific, the gamma function corresponding to each condition was downsampled to get 42 a_p values at equally spaced time points, and for each of that 42 values a simulation was performed keeping an $a=0.07$ for the pre-dose time interval and the a_p value after the drug administration. This procedure allowed to compute how the dynamics responded to the external perturbation at each a_p value over the extended period of time that was used to obtain the FCD. The rationale behind this procedure is to consider that the behavior of the system upon a periodic stimulation depends on its proximity to the bifurcation; therefore, by keeping this parameter constant and introducing the perturbation during an extended period of time, we can estimate how at any given time (equivalently, bifurcation parameter, as the a_p are linked to time by the optimal gamma function) the system would react to the perturbation.

Regarding the changes in the amplitude, F_{ext} , from 0 to 0.015 in 0.0125 steps, this was not increased with time. Instead, the procedure was repeated for each perturbation amplitude independently. Thus, for each perturbation amplitude we obtained an independent reactivity curve, with the aim of understanding how different perturbation intensities affected the temporal dynamics of each condition.

For further clarity, a pilot exploration with multiple perturbation amplitudes at different orders of magnitude was first conducted to determine the range where interesting effects could be found. After detecting that values above 0.015 resulted in flattened reactivity curves, we set that specific value as the maximum amplitude. The saturation of the correlation between 5HT2a receptor density and the reactivity as the amplitude of the perturbation is increased was a posterior and independent result,

showing that higher amplitudes not only flattened the reactivity curves, but the maxima of those curves also stopped correlating with the 5HT2a receptor density. As such, the correlation between the reactivity and the 5HT2a receptor density was certainly not used to determine the range of amplitudes of the perturbation, preventing the issue of circular reasoning.

To clarify this issue in the manuscript, we have changed the following passage in the “*Simulated perturbations*” subsection:

“To facilitate the interpretation of the results, we applied this perturbation to nodes located within six different resting state networks (RSN) identified using independent component analysis (ICA) by Beckmann, et al.. To account for the time variation of the reactivity, we sampled equally spaced values of $a(t)$, here noted a_p , with p indexing the time sample. In total we ended up with 42 a_p values corresponding to the gamma function of each condition sampled at these time points. Then, for every one of those values, we performed a simulation keeping a_p constant until the end of the simulation, i.e. for every simulation we set $a=0.07$ for the first 8 minutes, corresponding to the pre-dose time interval, and then keeping the constant value a_p until the end of the simulation. Therefore, the functional form of the bifurcation parameter is given by the concatenation of two constant functions, $a = 0.04$ and a_p , with $p = 0, \dots, 42$. This procedure allowed to compute how the dynamics responded to the external perturbation at each a_p value over the extended period of time that was used to obtain the FCD. Regarding the stimulation, we applied the perturbation after the first 8 minutes corresponding to the baseline, varying the amplitude F_{ext} from 0 to 0.015 in 0.0125 steps. The maximum value of this range was chosen as higher values saturated the local reactivity, flattening the curves. To summarize, for each combination of RSN, amplitude F_{ext} and the value of a_p , we computed the resulting FCD and its distance to the empirical condition, and then assessed the impact of the perturbation as explained in the following section.”

6. The used measure $\chi(t)$ of the sensitivity/reactivity is complicated and difficult to interpret. Additional explanations are necessary in this respect. I may also suggest that the reported results for sensitivity (Fig. 4) will be complemented by simple illustrations on how the FC matrices were affected over time by different stimulation intensities for DMT and placebo conditions starting from no-stimulation case.

We thank the reviewer for pointing out this issue. In order to clarify this explanation we re-formulated the subsection “*Measure of reactivity to perturbations*” as follows:

“We interpret the whole-brain model reactivity as the sensitivity of brain activity to changes in the external periodic stimulation. Following an analogy with the concept of susceptibility in statistical physics, we defined the reactivity as the following derivative:

$$\chi(t) = \frac{\partial M}{\partial F_{ext}}$$

where M denotes the Euclidean distance between the simulated and empirical FCD matrices. As F_{ext} is increased, we expect the stimulated FCD to depart from the baseline empirical value. $\chi(t)$ measures the rate at which this divergence occurs. Thus, a large $\chi(t)$ value indicates that at time t , the effect of changing the stimulation amplitude is maximal, measured in terms of its impact on the Euclidean distance between the simulated and empirical FCD matrices. Conversely, a small $\chi(t)$ represents a regime where changing the amplitude exerts comparatively little impact on the FCD. The reactivity was computed using a second order finite difference method. We evaluated $\chi(t)$ relative to its value at $t=0$ by subtracting the $\chi(t=0)$ at later times. This was done in order to capture the changes of the perturbation relative to the baseline part. Furthermore, given that the number of nodes differs across the RSNs, and that the reactivity can depend on the number of stimulated nodes, we normalized its value by the number of nodes of each RSN.”

Figure R10. Plots of the Functional Connectivity Dynamics (FCD) over time for 6 different time points corresponding to different bifurcation parameters drawn from the optimal gamma function for each condition. Results are shown for DMT and placebo, without perturbation (panel A) and with a perturbation amplitude of 0.01 applied at the visual RSN (panel B).

Regarding the suggestion of showing how the FCs are affected over time, we would like to remind the Reviewer that we are analyzing how the FCD is changing over time, not the FC. Nevertheless, following the suggestion made by the Reviewer, we have added a supplementary material plot showing how the FCD change over time selecting six different times post dose for the case of no

external stimulation (Supplementary Figure 7, reproduced here as Figure R10, panel A), and also in the case where an external perturbation of amplitude 0.01 was applied (Figure R10, panel B). Time = 0 indicates the time where the dose was administered. The figure corresponds to stimulation applied to the visual RSN. This new supplementary figure was cited in the Results section of the revised manuscript.

7. As mentioned in Ref. [9], "Values related to head motion (i.e., frame-wise displacement; FD) were significantly different between DMT and placebo", where 13 subjects were included in the dynamical analysis in contrast to 15 subjects in the current study. In that study two more subject sub-samplings were considered to address this problem, which is applicable also to the current manuscript. Can the motion artifacts affect the modeling results, or is the model little sensitive to them?

We thank the Reviewer for this observation. Indeed, given the different nature of the analyses conducted in both studies we opted to adopt a more inclusive criteria in our manuscript. Our analysis is primarily based on the FCD matrix, with entries reflecting the similarity between FC matrices at specific time points. While residual head motion artifacts can potentially introduce noise in the computation of FC matrices corresponding to specific time windows, these effects are limited to a minority of the FCD matrix elements. In comparison, the analyses conducted by Timmermann et al. include multimodal correlations computed over extended periods of time, where outliers can introduce significant bias to the final results of the reported analyses.

Reviewer #3

Major points:

Methods

- Psychedelics can have drastic effects on physiological effects (heart rate, blood pressure, etc.). Especially when it comes to functional connectivity analyses, these effects can induce spurious correlations between regions that are for example close to the same blood vessel. I noticed that the authors have followed the usual preprocessing steps for FC analyses, but do they have ongoing measures of heart rate and pulse during the fMRI scan which they could use to remove artifacts by explicitly modelling these artefacts (e.g. using this approach: <https://doi.org/10.1016/j.jneumeth.2016.10.019>). I believe that usual FC correction methods might not be good enough to account for these confounding factors. If this information is not available this should at least be clearly stated as a limitation.

We thank the Reviewer for raising this important discussion. As stated by the Reviewer, psychedelics exert significant physiological effects, especially affecting the cardiovascular system. In turn, these effects could result in confounds affecting FC estimates from resting state fMRI data. The preprocessing steps adopted in our work are standard in the field, in particular, they have been

adopted in the majority of fMRI studies of serotonergic psychedelics, including DMT^{42 43 44 45 46 47}. In the revised version of our manuscript, we have stated the potential limitations associated with indirect methods to estimate and remove physiological noise sources. We also expanded the discussion on the issue of altered neurovascular coupling due to the effects of psychedelics, which constitutes an important future area of research to improve current neuroimaging standards in the field.

- Could the authors provide a justification for using an anatomical parcellation over a functional parcellation (e.g., Schaefer Atlas) considering that they are interested in functional connectivity?

We thank the Reviewer for this comment, and the opportunity to justify our choice of brain parcellation. This justification is based both on theoretical and practical considerations. Starting with the latter, the AAL anatomical parcellation is one of the most common choices used to reduce the dimensionality of resting state fMRI data, providing regions of interest defined from anatomical landmarks and annotated using standardized terminology. This choice is consistent with previous modeling studies, where inter-areal coupling was introduced in terms of SC computed between regions in the AAL atlas^{48 49 50 51 52}. This facilitates decisions such as the range of model parameters to be explored during optimization, motivating the range of values for parameters G and a during the pilot exploration.

Beyond these practical considerations, we note that there is no established ground truth regarding the adequacy of brain parcellations; instead, some parcellations may be better than others at certain

⁴² Mueller, F., Lenz, C., Dolder, P. C., Harder, S., Schmid, Y., Lang, U. E., ... & Borgwardt, S. (2017). Acute effects of LSD on amygdala activity during processing of fearful stimuli in healthy subjects. *Translational psychiatry*, 7(4), e1084-e1084.

⁴³ Schmidt, A., Müller, F., Lenz, C., Dolder, P. C., Schmid, Y., Zanchi, D., ... & Borgwardt, S. (2018). Acute LSD effects on response inhibition neural networks. *Psychological medicine*, 48(9), 1464-1473.

⁴⁴ Carhart-Harris, R. L., Muthukumaraswamy, S., Roseman, L., Kaelen, M., Droog, W., Murphy, K., ... & Nutt, D. J. (2016). Neural correlates of the LSD experience revealed by multimodal neuroimaging. *Proceedings of the National Academy of Sciences*, 113(17), 4853-4858.

⁴⁵ Carhart-Harris, R. L., Erritzoe, D., Williams, T., Stone, J. M., Reed, L. J., Colasanti, A., ... & Nutt, D. J. (2012). Neural correlates of the psychedelic state as determined by fMRI studies with psilocybin. *Proceedings of the National Academy of Sciences*, 109(6), 2138-2143.

⁴⁶ Grimm, O., Kraehenmann, R., Preller, K. H., Seifritz, E., & Vollenweider, F. X. (2018). Psilocybin modulates functional connectivity of the amygdala during emotional face discrimination. *European Neuropsychopharmacology*, 28(6), 691-700.

⁴⁷ Timmermann, C., Roseman, L., Haridas, S., Rosas, F. E., Luan, L., Kettner, H., ... & Carhart-Harris, R. L. (2023). Human brain effects of DMT assessed via EEG-fMRI. *Proceedings of the National Academy of Sciences*, 120(13), e2218949120.

⁴⁸ Ipiña, I. P., Kehoe, P. D., Kringelbach, M., Laufs, H., Ibañez, A., Deco, G., ... & Tagliazucchi, E. (2020). Modeling regional changes in dynamic stability during sleep and wakefulness. *NeuroImage*, 215, 116833.

⁴⁹ Sanz Perl, Y., Pallavicini, C., Pérez Ipiña, I., Demertzi, A., Bonhomme, V., Martial, C., ... & Tagliazucchi, E. (2021). Perturbations in dynamical models of whole-brain activity dissociate between the level and stability of consciousness. *PLoS computational biology*, 17(7), e1009139.

⁵⁰ Clusella, P., Deco, G., Kringelbach, M. L., Ruffini, G., & Garcia-Ojalvo, J. (2023). Complex spatiotemporal oscillations emerge from transverse instabilities in large-scale brain networks. *PLOS Computational Biology*, 19(4), e1010781.

⁵¹ Deco, G., Cruzat, J., & Kringelbach, M. L. (2019). Brain songs framework used for discovering the relevant timescale of the human brain. *Nature communications*, 10(1), 583.

⁵² Luppi, A. I., Craig, M. M., Pappas, I., Finoia, P., Williams, G. B., Allanson, J., ... & Stamatakis, E. A. (2019). Consciousness-specific dynamic interactions of brain integration and functional diversity. *Nature communications*, 10(1), 4616.

applications^{53 54}. While a functional parcellation may seem an adequate study for fMRI-based analysis, there are reasons to support the use of the AAL parcellation instead. This parcellation is based on anatomical landmarks which may be better aligned with cytoarchitectonic information. In turn, the differential expression of 5HT2a receptors throughout the human brain may be better aligned with the divisions introduced by cytoarchitectonic characterization in contrast to those arising from functional data. This is supported by studies showing that 5HT2a receptors are preferential to cells with pyramidal morphology⁵⁵. Importantly, fMRI studies suggest that psychedelic drugs exert anatomically heterogeneous effects of BOLD signals, which are better understood in terms of 5HT2a receptor density relative to functional characterizations of the cortex, for instance, in terms of Resting State Networks (RSN). As a concrete example, the study by Tagliazucchi et al. shows that the voxel-wise density of functional connections increases under the acute effects of LSD, and that the outline of these changes shows higher correlation with the spatial profile of 5HT2a receptor distribution than with the canonical RSN⁵⁶. While future studies should address the suitability of function vs. anatomical/cytoarchitectonic parcellations to describe the neural effects elicited by serotonergic psychedelics, we believe both approaches have sufficient merit to be adopted given the current knowledge.

A final relevant point concerns the compatibility of functional parcellations with the structural connectivity. Thus, the reasoning that could motivate the use of the Schafer parcellation to compute functional connections applies to the estimation of anatomical connections using an anatomical parcellation, such as the AAL. In other words, it is not clear that a functional parcellation divides the brain into regions whose structural connections can be optimally measured using DTI. Overall, we believe that the potential benefits of the Schafer parcellation do not outweigh the concrete and practical advantages provided by the AAL parcellation in the context of the present work.

- When it comes to the modelling, I am not convinced that modelling the placebo response with a gamma function makes sense. This choice seems reasonable for the DMT condition, but less well suited for placebo in the absence of a pharmacological intervention. Indeed eyeballing Figure 4, the gamma function in the placebo condition appears to approximate a constant effect. Could the authors model the time-varying effect as a constant term + a gamma function that is only active in the DMT condition? Perhaps the authors could justify this choice, or consider changing their model for the placebo condition. Alternatively, I would like to see a comparison with a null-model with a constant a parameter. Presumably, this model would perform better or as good in the placebo condition, but worse in the LSD condition, which would be informative to show.

We thank the Reviewer for this interesting observation. Indeed, for a wide range of parameter values, the gamma function is not expected to reproduce the optimal temporal evolution of $a(t)$ for the placebo condition. However, this function becomes increasingly similar to the constant null function as β increases and λ decreases. *A priori*, we hypothesized that the optimal fit would result in $a(t)$ paralleling the typical shape drug effect intensity for DMT, and in a null constant function for the

⁵³ Moghimi, P., Dang, A. T., Do, Q., Netoff, T. I., Lim, K. O., & Atluri, G. (2022). Evaluation of functional MRI-based human brain parcellation: a review. *Journal of neurophysiology*, 128(1), 197-217.

⁵⁴ Eickhoff, S. B., Yeo, B. T., & Genon, S. (2018). Imaging-based parcellations of the human brain. *Nature Reviews Neuroscience*, 19(11), 672-686.

⁵⁵ Willins, D. L., Deutch, A. Y., & Roth, B. L. (1997). Serotonin 5-HT2A receptors are expressed on pyramidal cells and interneurons in the rat cortex. *Synapse*, 27(1), 79-82.

⁵⁶ Tagliazucchi, E., Roseman, L., Kaelen, M., Orban, C., Muthukumaraswamy, S. D., Murphy, K., ... & Carhart-Harris, R. (2016). Increased global functional connectivity correlates with LSD-induced ego dissolution. *Current biology*, 26(8), 1043-1050.

placebo condition. To address this hypothesis while performing a fair comparison between both conditions, we employed the same function (i.e. gamma function); however, we expected that model optimization would result in large β and small λ for the placebo condition, i.e. that we would obtain the limiting case corresponding to the null constant function. In other words: the Reviewer suggested that the placebo condition is better modeled by an approximately constant $a(t)$, and we note that this behavior is a particular case of the more general behavior described by the gamma function, which is indeed obtained when fitting the model to the data. This confirms both our hypothesis and the Reviewer's intuition.

Nevertheless, as suggested by the Reviewer, we incorporated the analysis of constant $a(t)$ as a null model. The left panel of supplementary Figure 8 (reproduced here as Figure R11) presents the distribution of Euclidean distance between empirical and simulated FCD for 50 simulations, both for DMT and placebo. Performing a Kolmogorov-Smirnov test (KS test) shows that there are no differences between the distributions in either of the conditions, using a threshold of 0.01 for statistical significance. However, if we only consider the matrix elements of the FCD corresponding to the post-dose interval (where the dynamics introduced by the DMT infusion are manifest), we found that the constant $a(t)$ outperforms the gamma function for the placebo (Figure R12B). While this supports the lack of significant dynamics post-infusion for the placebo, we nevertheless maintained our comparison using the gamma function based on the justification provided in the above paragraph.

Figure R11. The left panel compares the minima for the null-model (uniform function) and the gamma model when analyzing the entire FCD. The right panel compares the minima for the null-model (uniform function) and the gamma model when comparing only the elements of FCDs post dose. Kolmogorov Smirnov test show no difference between models for the entire FCD in either condition. For the post-dose FCD the uniform model has shown lower values for the placebo (PCB) condition.. No difference was found for the DMT.

- Figure 4: I am not quite sure, what the advantage of estimating the gamma function from fMRI data compared to using the known pharmacogenetics of DMT is. Is this just to demonstrate that it can be estimated from this data? Could the authors clarify what the advantage of estimating it from the data is. Perhaps the authors could also assess the correlation between a canonical pharmacogenetics function and their estimated functions.

We thank the Reviewer for this question, which is important to convey the relevance of our work. The main goal of our study was not to estimate the pharmacokinetics of the DMT, as there are other methods better suited for this objective. Because of this, we adopted a phenomenological simplified

model that provides a conceptual characterization of the transient dynamics induced by DMT. Instead, our work establishes that changes in dynamic FC post-infusion parallel the characteristic shape of DMT pharmacokinetics. Moreover, the model also shows that this temporal evolution is consistent with a shifting of the proximity to the bifurcation point, and therefore of the stability of the dynamics. As such, our work not only described dynamic FC in terms of an intrinsic pharmacological property of the psychedelic compound, but also suggested a dynamical mechanism underlying this description. Methodologically, showing that DMT pharmacokinetics underlies dynamic FC is non-trivial, given that the non-stationary component of the BOLD signals which may reflect this process are removed or attenuated as part of standard pre-processing. Thus, DMT pharmacokinetic cannot be directly reflected in the BOLD signals themselves, as this component is intertwined with low frequency artifacts that are removed by a high pass filter. Our work shows that the high pass component of the signals is sufficient to introduce a temporal evolution of the FC which parallels the drug pharmacokinetics.

Besides the use of a whole-brain model to provide this mechanistic characterization, we believe our approach carries sufficient methodological novelty to merit its application to other datasets and conditions. To date, the vast majority of whole-brain modeling studies is concerned with estimating parameters to reproduce observables computed from resting state recordings, such as the structure of the FC matrix, or the statistical distribution of FCD matrix entries. In contrast, our approach could be applicable to investigate transients introduced by other forms of stimulation and/or cognitively demanding tasks, as well as other physiological or pathological paroxysmic events (e.g. sleep graphoelements, abnormal events and transient in epileptic patients, etc.). This requires the temporal parametrization of a set of model variables, following one or more hypotheses concerning the time course of the phenomenon under study. The model fit can be used to contrast between hypotheses and/or conditions, while simultaneously addressing potential mechanisms in terms of the variables reflected modulated in time. To the best of our knowledge, this form of time-resolved whole-brain modeling is a novel introduction of our manuscript.

- Figure 4B: Reactivity depends on stimulation intensity. Could the authors explain why stronger perturbations lead to less reactivity for the EC network. In the discussion (p.15, ll. 326ff) the authors argue: “For high values of the external perturbation amplitude, the reactivity decoupled from the 5HT2a receptor density, likely indicating the saturation of the induced effects on whole-brain dynamics.” This explanation is plausible for networks where the increasing stimulation leads to a plateau in reactivity (maybe Aud, ES and SM), but does this really explain why the reactivity goes down in EC? Perhaps the authors could comment on this.

In agreement with the Reviewer, we note that the peak reactivity decreased for the larger stimulation amplitudes in the case of the EC network (Figure 4B of the main manuscript text). We first note that the reactivity is defined as a derivative, measuring how much the Euclidean distance between stimulated and non-stimulated FCD changes as a function of the stimulation amplitude. Thus, this decrease means that for larger stimulation amplitudes, the effect of increasing the amplitude diminishes or stabilizes. Asymptotically, this is to be expected, as the stimulated FCD will eventually converge to a certain fixed configuration for very large stimulation amplitudes, which will stabilize the derivative close to zero. This is consistent with the behavior showcased in Figure 4B, where several RSN show an initial increase of the reactivity, followed by a decrease. For the EC network, this decrease brings the reactivity closer to zero than for other RSN. However, as discussed above, this behavior is eventually expected to occur for all RSN. The reasons why the EC network presents this behavior earlier than other networks may be complex, involving both the spatial extent of the network

and the detailed topology of the involved network nodes. Therefore, we consider that their detailed investigation represents a significant detour from the main objective of our work, ideally postponed for a future study focused on this and related topics.

Concerning the decoupling between the reactivity and the 5HT2a density map, indeed we observed that for large stimulation amplitudes, the regional correlation between these two maps was diminished. It is reasonable to assume that sufficiently large stimulation amplitudes eventually drive the FCD to asymptotic stable configuration regardless of the RSN being stimulated, leading to decreased and regionally homogeneous reactivity values. Due to this spatial homogenization we observe a drop in the correlation coefficient ρ for large stimulation amplitudes. The plateau in networks such as Aud, ES and SM indicates the beginning of this process, which occurs earlier (i.e. for a smaller stimulation amplitude) for the EC network, but in all cases results in decrease of the correlation between 5HT2a anatomical variation and the regional reactivity due to the saturation and spatial homogenization of the latter. We note that this does not explain why the reactivity goes down in the EC network, as it is the opposite situation, i.e. the drop in ρ can be understood as a consequence of the stabilization of the reactivity.

2nd response to the Reviewers of Communications Biology COMMSBIO-24-1476, "Transient destabilization of whole brain dynamics induced by DMT"

We thank the reviewers for carefully reading and approving our responses. In the document below, we address the final concern raised by **Reviewer #3**. (Reviewer comment is in red).

In my initial review I asked the authors whether a constant $a(t)$ would be the better model for the placebo condition. The authors now included a comparison with this null model in the supplement, which I greatly appreciated. The authors say that the constant model significantly outperforms the gamma model in the placebo condition as I had suspected, but none of the other comparisons are significantly different. I am a bit surprised that they do not find a significant difference in favor of the gamma model in the DMT condition. Doesn't this suggest that a constant model would be as good in explaining the data as the gamma model even in the DMT condition, which would be a major challenge to some of the conclusions the authors draw? Could the authors comment on this? I also did not find a reference to this figure/analysis in the main manuscript. Could the authors add that to the main manuscript.

We thank the Reviewer for this comment. Indeed, we did not observe significant differences in the goodness of fit between the constant $a(t)$ and the gamma function model. However, we note that within the family of gamma functions used to parametrize the bifurcation parameter, the optimal fit was consistently obtained in a region of parameter space corresponding to a relatively early peak at ≈ 10 min, and a peak value that was in all cases larger than the peak value obtained for the placebo condition. These parameter values are compatible with human studies of DMT pharmacokinetics (for example, see Fig. 1 of Good et al., 2023¹). In other words, the parameter space included an ample variety of parametrizations for $a(t)$ including, in the limiting case of small β and large λ , a model tending towards a constant function, which was the optimal model for the placebo; however, fitting the gamma function model to the DMT data yielded parameter values consistent with experimental data.

As correctly stated by the Reviewer, this parametrization of $a(t)$ did not supersede the constant model in terms of the goodness of fit. We believe this result could depend on several factors, for instance, the choice of optimization metric and the relative weight given to relative differences between matrix values vs. the overall magnitude of the entries. As this was a first effort to fit whole-brain models to fMRI recordings of a transient brain state, we adopted relatively standard and/or straightforward methods to fit the model to empirical FC matrices; however, future efforts could systematically explore the possible methodological alternatives.

In agreement with the Reviewer's recommendation, we have commented on this issue on the main manuscript text. For this purpose, we modified the following paragraph of the "Discussion" section as follows:

*"However, since we did not specify a priori the direction of the effect, this approach allowed us to test whether DMT could bring the dynamics closer to the global bifurcation and therefore towards a point of decreased stability. **When comparing the gamma function with a null model keeping $a(t)$***

¹ Good, M., Joel, Z., Benway, T., Routledge, C., Timmermann, C., Erritzoe, D., ... & James, E. (2023). Pharmacokinetics of N, N-dimethyltryptamine in humans. *European Journal of Drug Metabolism and Pharmacokinetics*, 48(3), 311-327.

constant post-dose, we found that this parametrization outperformed the gamma function for the placebo condition, as expected from our results. However, we did not find significant differences relative to the gamma function for the DMT condition (see supplementary Figure 3). We note that within the family of gamma functions used to parametrize the bifurcation parameter, the optimal fit was consistently obtained in a region of parameter space corresponding to a relatively early peak at ≈ 10 min, and a peak value that was in all cases larger than the peak value obtained for the placebo condition. These parameter values are compatible with human studies of DMT pharmacokinetics (Good et al., 2023). The lack of difference observed between both models could depend on methodological choices (e.g. the choice of the optimization metric and the need to balance relative differences between matrix values vs. the overall magnitude of the matrix entries), which could be addressed in future studies fitting whole-brain models to transient brain states.”